# Value representations in the rodent orbitofrontal cortex drive learning, not choice

Kevin J Miller[1,2,3]*, Matthew M Botvinick[2,4]*, Carlos D Brody[1,5]*

[1]Princeton Neuroscience Institute, Princeton University, Princeton, United States; [2]DeepMind, London, United Kingdom; [3]Department of Ophthalmology, University College London, London, United Kingdom; [4]Gatsby Computational Neuroscience Unit, University College London, London, United Kingdom; [5]Howard Hughes Medical Institute and Department of Molecular Biology, Princeton University, Princeton, United States

**Abstract** Humans and animals make predictions about the rewards they expect to receive in different situations. In formal models of behavior, these predictions are known as value representations, and they play two very different roles. Firstly, they drive *choice*: the expected values of available options are compared to one another, and the best option is selected. Secondly, they support *learning*: expected values are compared to rewards actually received, and future expectations are updated accordingly. Whether these different functions are mediated by different neural representations remains an open question. Here, we employ a recently developed multi-step task for rats that computationally separates learning from choosing. We investigate the role of value representations in the rodent orbitofrontal cortex, a key structure for value-based cognition. Electrophysiological recordings and optogenetic perturbations indicate that these representations do not directly drive choice. Instead, they signal expected reward information to a learning process elsewhere in the brain that updates choice mechanisms.

*For correspondence:
kevinjmiller@deepmind.com
(KJM);
botvinick@deepmind.com
(MMB);
brody@princeton.edu (CDB)

**Competing interest:** The authors declare that no competing interests exist.

## Editor's evaluation

In this manuscript, Miller et al., use the two-step task, a task initially designed to discern model-free from model-based behavior, to probe OFC at specific times throughout the two stages of the task to understand OFC's role in choice and learning. The authors exploited an interesting feature of the two-step task that allows choice and learning to be separated into separate stages. They then used this feature to clearly show that OFC is not necessary for choice behavior in a well-learned choice task, although it is required for updating the model based on trial-by-trial reward information. The ability to separate learning from choice in a decision-making task is a unique and novel approach for probing OFC function, making this manuscript an important contribution to our understanding of OFC function.

## Introduction

Representations of expected value play a critical role in human and animal cognition (*Sugrue et al., 2005*; *Lee et al., 2012*; *Daw and O'Doherty, 2014*). A key brain region involved in representing and using expected value information is the orbitofrontal cortex (OFC). An active and unresolved debate in the literature involves theoretical accounts of the OFC that emphasize roles in either choosing (*Wallis, 2007*; *Padoa-Schioppa and Conen, 2017*), in learning (*Schoenbaum et al., 2009*; *Walton et al.,*

2011; *Song et al., 2017*), or both (*O'Doherty, 2007*; *Wallis, 2012*; *Rudebeck and Murray, 2014*; *Wilson et al., 2014*; *Stalnaker et al., 2015*). Studies perturbing OFC activity have variously reported behavioral effects consistent with either altered learning (*Takahashi et al., 2009*; *Walton et al., 2010*; *McDannald et al., 2011*; *Jones et al., 2012*; *Gardner et al., 2019*; *Gardner et al., 2020*), or with altered choosing, including conditions involving choosing between multiple possible actions (*Murray et al., 2015*; *Ballesta et al., 2020*; *Kuwabara et al., 2020*) and involving choosing the frequency with which to perform a single action (*Jones et al., 2012*; *Gremel and Costa, 2013*). Recording studies in many species have revealed neural correlates of expected value in the OFC (*Thorpe et al., 1983*; *Schoenbaum et al., 1998*; *Gottfried et al., 2003*; *Padoa-Schioppa and Assad, 2006*; *Sul et al., 2010*), but it has not been clear whether these neural correlates of expected value are selective for roles in learning and in choosing. This, along with limitations inherent in neural perturbation studies, has made it difficult to determine whether the OFC plays a role in learning, choosing, or both.

Perturbation studies designed to separate learning from choosing have typically adopted a temporal strategy, attempting to behaviorally separate the time of learning from the time of choosing, and performing neural silencing experiments specifically at one time or the other. Some experiments of this type are consistent with a role for the OFC in learning: silencing the OFC specifically during the learning phase of either of two associative learning procedures (blocking or overexpectation) impairs subsequent behavior on an assay performed with the OFC intact (*Takahashi et al., 2009*; *Jones et al., 2012*). Other experiments of this type are consistent with a role for the OFC in choosing: silencing the OFC specifically during the post-learning assay of either of two associative learning procedures (sensory preconditioning or outcome devaluation) impairs expression of knowledge gained earlier (*Jones et al., 2012*; *Gremel and Costa, 2013*; *Murray et al., 2015*).

One limitation of studies that attempt to separate learning and choosing temporally is that these are fundamentally cognitive events, internal to the brain, and their timing cannot be fully controlled by an experimenter. For example, it has been suggested that subjects faced with a choice might reconsider information that was presented in the past; cognitively, this might best be interpreted as a learning process happening during a putative choice epoch (*Gershman et al., 2014*; *Lombrozo, 2017*; *Ludvig et al., 2017*). Conversely, subjects experiencing a surprising outcome might consider its implications for future choices that they expect to be faced with, and memorize decisions for later use (*McDaniel and Einstein, 2007*) cognitively, this might best be interpreted as a choice process happening during a putative learning epoch. There is no a priori requirement for learning and choosing to be implemented in separate neural circuits. The approach we used here does not depend only on timing, but importantly also uses signals internal to the brain (firing rates, effect of OFC perturbations), to quantify the relative contributions of learning and choosing in the OFC. A second limitation is of previous studies is that they typically address forms of learning that unfold over many sessions, a very different timescale and behavioral regime to the one in which neural representations of value in the OFC are typically studied. Since different neural mechanisms may be at play in trial-by-trial vs. session-by-session learning, it is difficult to confidently interpret the results of neural recording experiments addressing the former in light of neural silencing experiments addressing the latter.

Studies characterizing neural correlates of expected value in the OFC have used a different set of behavioral methods. Typically, they adopt a task design which facilitates the analysis of neural recording by allowing data to be aggregated over many repeated trials. In each trial, the subject is presented with one or more options, chooses among them, and receives a reward associated with the chosen option. Some studies (*Thorpe et al., 1983*; *Wallis and Miller, 2003*; *Sul et al., 2010*; *Costa and Averbeck, 2020*) present options that are novel or have repeatedly changing reward associations in order to elicit trial-by-trial learning, interleaving decision-making with learning by design. Other studies (*Tremblay and Schultz, 1999*; *Padoa-Schioppa and Assad, 2006*; *Kennerley et al., 2009*; *Rudebeck et al., 2013*; *Blanchard et al., 2015*; *Constantinople et al., 2019*) present options whose reward associations are stable and well-learned, with the intention of isolating decision-making specifically. Although not incentivized by task design, trial-by-trial learning is frequently evident in these tasks as well (*Padoa-Schioppa, 2013*; *Constantinople et al., 2019*; *Lak et al., 2020*). The fact that learning and choosing are intertwined in these tasks, operating at similar times and over the same set of items, has made it difficult to determine whether OFC value representations are selective for one or the other process, or whether they play a role in both.

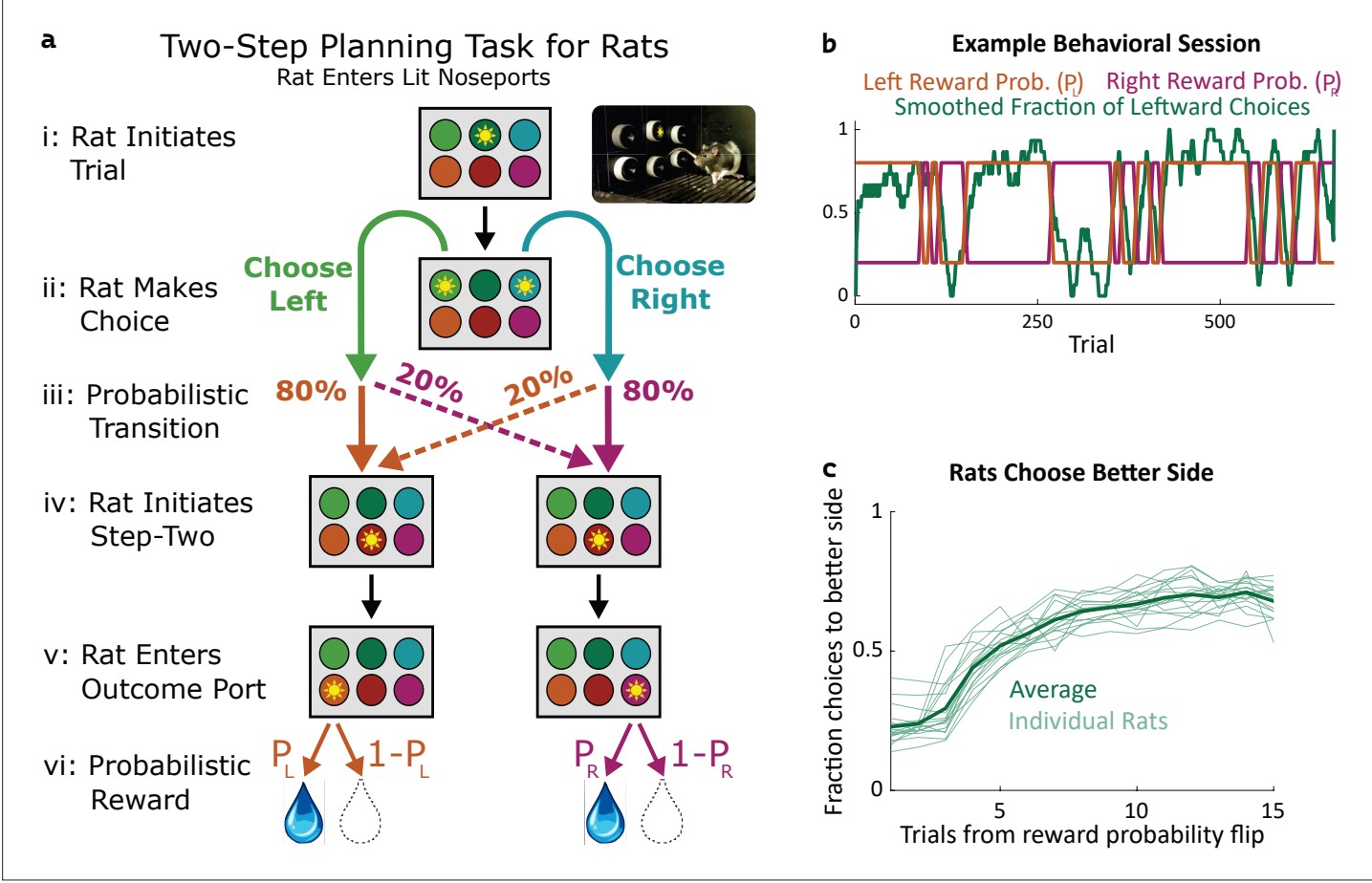

**Figure 1.** Two-step task for rats. (**a**) Rat two-step task. The rat initiates a trial by entering the top center port (**i**), then chooses to enter one of two choice ports (**ii**). This leads to a probabilistic transition (**iii**) to one of two possible paths. In both paths, the rat enters the bottom center port (**v**), causing one of two outcome ports to illuminate. The rat enters that outcome port (**v**), and receives a reward (**vi**). (**b**) Example behavioral session. At unpredictable intervals, outcome port reward probabilities flip synchronously between high (80%) and low (20%). The rat adjusts choices accordingly. (**c**) The fraction of trials on which each rat (n=19) selected the choice port whose common (80%) transition led to the outcome port with the currently higher reward probability, as a function of the number of trials that have elapsed since the last reward probability flip.

Here, we adopt a computational approach that separates learning from choosing, even while using a repeated-trials task that facilitates analysis of neural recordings. This approach is based not only on *when* learning versus choosing happens, but also on *what content* they operate over. We use a multi-step decision task *Daw et al., 2011*; *Miller et al., 2017*, in which a choice made on a first step is linked probabilistically to an outcome that occurs on a second step, which in turn is linked probabilistically to reward (*Figure 1a*). Rats adopt a model-based planning strategy that respects this structure (*Figure 2a*). On the first step of every trial rats make their choice based on values that are computed, not learned ('Compute Choice Port Values' and 'Choose a Choice Port' in *Figure 2a*), while on the second step they learn the values of the outcomes ('Learn Outcome Port Values' in *Figure 2a*) but are led to those outcomes without making a choice. This task structure, in which choices and outcomes are linked probabilistically rather than having a 1-to-1 relationship, combined with the planning strategy rats use to solve it, provides the critical separation necessary to differentiate learning from choosing.

We asked whether OFC neuron firing rates were correlated with the expected values of the items being learned about, the values being chosen between, or both. We find that neurons in the OFC correlate significantly with the expected values of the items being learned about, but only weakly with the expected values of the items being chosen between. This indicates that, within a repeated-trials task, neural representations in the OFC carry value information that is largely selective for a role in learning, but only weakly carry value information selective for a role in choice. To causally probe whether OFC plays a role in learning, in choosing, or in both processes, we transiently silenced

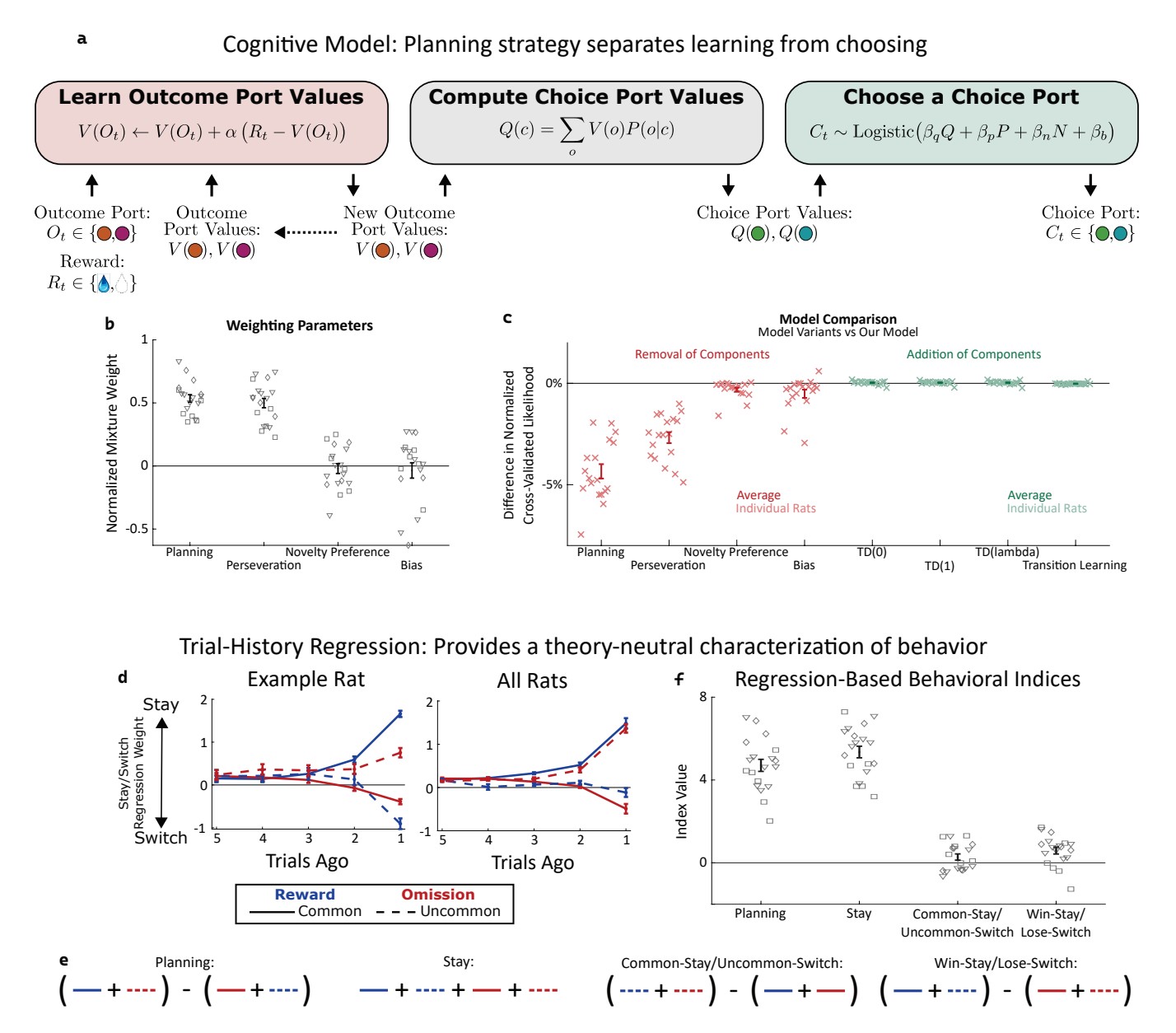

**Figure 2.** Planning strategy separates learning, choosing. (**a**) Schematic of the planning strategy. Agent maintains value estimates (**V**) for each outcome port, based on a history of recent rewards at that port; as well as value estimates (**Q**) for each choice port, which are computed on each trial based on the outcome values and the world model (*P(o|c)*). Choices are drawn probabilistically, based on a weighted combination of these values and of the influence of three other behavioral patterns: perseveration, novelty preference, and bias (see Methods for details). (**b**): Mixture weights of the different components of the cognitive model fit to rats' behavioral data. shown for electrophysiology rats (n=6, squares), optogenetics rats (n=9, triangles), and sham optogenetics rats (n=4, diamonds). (**c**) Change in quality of model fit resulting from removing components from the model (red) or adding additional components (green). (**d**) Fit weights of the trial-history regression for an example rat (left) and averaged over all rats (right). (**e**) Definitions of the four behavioral indices in terms of the fit stay/switch regression weights. The planning index for a particular rat is defined as the sum of that rat's common-reward and the uncommon-omission weights, minus the sum of its common-omission and uncommon-reward weights. The 'stay' index is defined as the sum of all weights. (**f**) Values of the four behavioral indices for all rats. The planning and stay indices are large and positive for all rats, while the common-stay/uncommon-switch and win-stay/lose-switch indices are smaller and inconsistent in sign.

The online version of this article includes the following figure supplement(s) for figure 2:

**Figure supplement 1.** Learning rate parameters.

**Figure supplement 2.** Weights of trial-history regression model, fit both to rats' behavioral data and to synthetic datasets generated by our mixture-of-agents model fit separately to each rat.

OFC activity, and found that the pattern of behavior induced by this silencing was reproduced in our computational model only when we disrupted only the role of expected value in learning. Disrupting the role of value in choice did not reproduce this effect. In this model, learning could still occur, but was impaired due to one of its key input signals, expected value, being disrupted.

These results suggest that, even when choosing and learning are commingled on the same trial-by-trial timescale, value representations in the rodent OFC do not drive choices directly, but instead support learning. Moreover, we identify a specific computational role for OFC within learning: it supplies information about expected outcomes to a separate learning process, elsewhere in the brain, that updates value estimates, which can then in turn be used to drive future behavior (*Schoenbaum et al., 2011*).

## Results

### Planning strategy in the two-step task separates choosing and learning

We trained rats to perform a multistep decision-making task in which they adopt a strategy of model-based planning (*Miller et al., 2017*). The structure of the task was as follows: The rat initiated each trial by poking its nose into a neutral center port, and then selected one of two choice ports (*Figure 1a* **i,ii**). One choice caused a left outcome port to become available with probability 80% ('common' transition), and a right outcome port to become available with probability 20% ('uncommon' transition), while the opposite choice reversed these probabilities (*Figure 1a* **iii**). These transition probabilities were fixed for each rat, but counterbalanced across rats. Following the initial choice, an auditory cue informed the rat which of the two outcome ports had in fact become available on that trial and, after poking into a second neutral center port, the available outcome port was further indicated by a light (*Figure 1a* **iv,v**). The rat was required to poke into the available outcome port (no choice in this step), where it received a water reward with some probability (*Figure 1a* **v,vi**). The reward probability at each outcome port on each trial was either 80% or 20%, and these probabilities reversed at unpredictable intervals (*Figure 1b*). Earning a high reward rate on this task requires learning which outcome port currently has the higher reward probability, and choosing the choice port that is more likely to lead to that outcome port. Rats did this successfully, switching their average choices to the appropriate choice port within several trials of a reward probability reversal (*Figure 1b and c*).

A high reward rate on this task can be achieved by multiple cognitive strategies *Daw et al., 2011*; *Kool et al., 2016*. In previous work (*Miller et al., 2017*), we showed that rats solve the task using a particular strategy termed 'model-based planning' (*Dolan and Dayan, 2013*; *Miller and Venditto, 2021*), and presented a cognitive model implementing this strategy (*Figure 2a*). Model-based planning in our task involves separate representations of the expected value associated with the outcome ports and of the expected value associated with the choice ports, and very different computational roles for each type of value representation. Outcome port values, (labeled *V(o)* in our model), represent an estimate of the reward that can be expected following a visit to the corresponding outcome port (*o*). They are updated incrementally by a learning process that compares this expectation to the reward actually received on each trial (labeled $R_t$, *Figure 2a*, left: 'Learn Outcome Port Values'). We used a symmetric update for the value of the outcome port that was not visited (see Methods). Choice port values (labeled *Q(c)* in our model) represent an estimate of the reward that can be expected following a visit to the corresponding choice port (*c*). They are computed based on the known transition probabilities between first-step choice and second-step outcome (the 'world model', *P(o|c)*; *Figure 2a*, center: 'Compute Choice Port Values'). These choice port values are then used to determine the next choice (*Figure 2a*, right: 'Choose a Choice Port'). The values of the choice ports therefore drive choice directly, while the learned values of the outcome ports support choice only indirectly, by directly supporting learning.

As in our previous study (*Miller et al., 2017*), rat behavior was well-described by a cognitive model combining this model-based planning strategy with a mixture of three additional components. The first of these is 'perseveration', which reflects a tendency to repeat past choices, regardless of their outcomes (*Akaishi et al., 2014*; *Miller et al., 2019*). The second, which we term 'novelty preference', reflects a tendency to repeat (or to switch away from) choices that lead to an uncommon transition, regardless of whether or not they are rewarded. The third is a constant side bias, reflecting an overall tendency to prefer either the right or the left choice port. Each of these components is associated with

a weighting parameter (β), reflecting the strength of its influence on the decision between the left and right choice port on each trial (*Figure 2a*, right: 'Choose a choice port'). Fitting these parameters to the dataset for each rat, we find that the planning and perseverative components earn weights that are large and positive (*Figure 2b*), while the novelty preference and bias components earn weights that are generally smaller and differ in sign among rats (*Figure 2b*). The planning and perseveration components are each associated with a learning rate parameter (α), which reflects the relative influence of trials in the recent past vs the more distant past. Learning rates for the planning component were consistently larger than those for the perseverative component (*Figure 2—figure supplement 1*).

We validate that our cognitive model provides a good description of rat behavior in two different ways. The first way is quantitative model comparison: we compute a quality-of-fit score for the model using cross-validated likelihood, and compare this score between our model and various alternatives (*Figure 2c*). Alternative models which are missing any of the four components perform substantially worse (*Figure 2c*, red points), while alternative models adding various additional components do not perform substantially better (*Figure 2c*, green points). Among these additional components we tested were several model-free reinforcement learning strategies, which have been reported to contribute to behavior on similar multi-step tasks *Daw et al., 2011*; *Hasz and Redish, 2018*; *Dezfouli and Balleine, 2019*; *Groman et al., 2019*; *Akam et al., 2020*; *Miranda et al., 2020*. That adding them to our model does not improve quality of fit suggests that model-free reinforcement learning does not contribute meaningfully to rat behavior in our task. This is important for our analysis because these strategies involve value representations of their own – that they are not part of our model means that model-based value representations are the only representations of expected reward that are present.

Our second way of validating the cognitive model makes use of a trial-history regression analysis (*Lau and Glimcher, 2005*), which provides a theory-neutral way of characterizing the patterns present in a behavioral dataset. This analysis fits separate weights for each of the four possible outcome types (common-rewarded, common-omission, uncommon-rewarded, uncommon-omission). These weights will be positive if the rat tends to repeat choices that lead to that outcome, and negative if the rat tends to switch away from such choices. Behavioral datasets produced by different strategies will have different patterns of weights (*Miller et al., 2016*). For example, a model-based planning strategy will show more positive weights for common-reward than for uncommon-reward (since a rewarding outcome port visited after a common transition is likely to be reached again by repeating the choice, while one reached after an uncommon transition is more likely to be reached by switching to the other choice port instead), as well as more negative weights for common-omission than for uncommon-omission (since a unrewarding outcome port visited after a common transition is likely to be avoided by switching the choice, while one reached after an uncommon transition is more likely to be avoided by repeating the choice). As in our previous study, rats trained for the present study universally show this qualitative pattern, and the quantitative patterns present in their weights are well-matched by fits of the cognitive model (example rat: *Figure 2d*; all rats: *Figure 2—figure supplement 2*). To summarize the pattern and compare it to others in behavior, we define a 'planning index' as the linear combination of weights consistent with the planning strategy, as well as a 'stay' index, a 'common-stay/uncommon-switch' index, and a 'win-stay/lose-switch' index quantifying other patterns (*Figure 2e*). All rats showed large values of the planning index and the stay index, and much smaller values of the common-stay/uncommon-switch and win-stay/lose-switch indices (*Figure 2f*).

This cognitive model allows us to probe the role of the OFC in two key ways. First, the model can be run using choices and rewards actually experienced by the rat to provide trial-by-trial timecourses for the expected values of the choice ports (Q) and outcome ports (V). We will use these timecourses (*Daw, 2011*) as estimates of the value placed by the rat on the various ports, and look for correlates of them in the activity of OFC neurons. Second, the model can be altered selectively and run to generate synthetic behavioral datasets, providing predictions about the specific behavioral effects of particular cognitive impairments. We will selectively disrupt choice port value information (which directly drives choice) or outcome port value information (which participates in learning) within the model on a random subset of trials. We will compare these synthetic datasets to real behavioral datasets from rats in which we silence neural activity in the OFC.

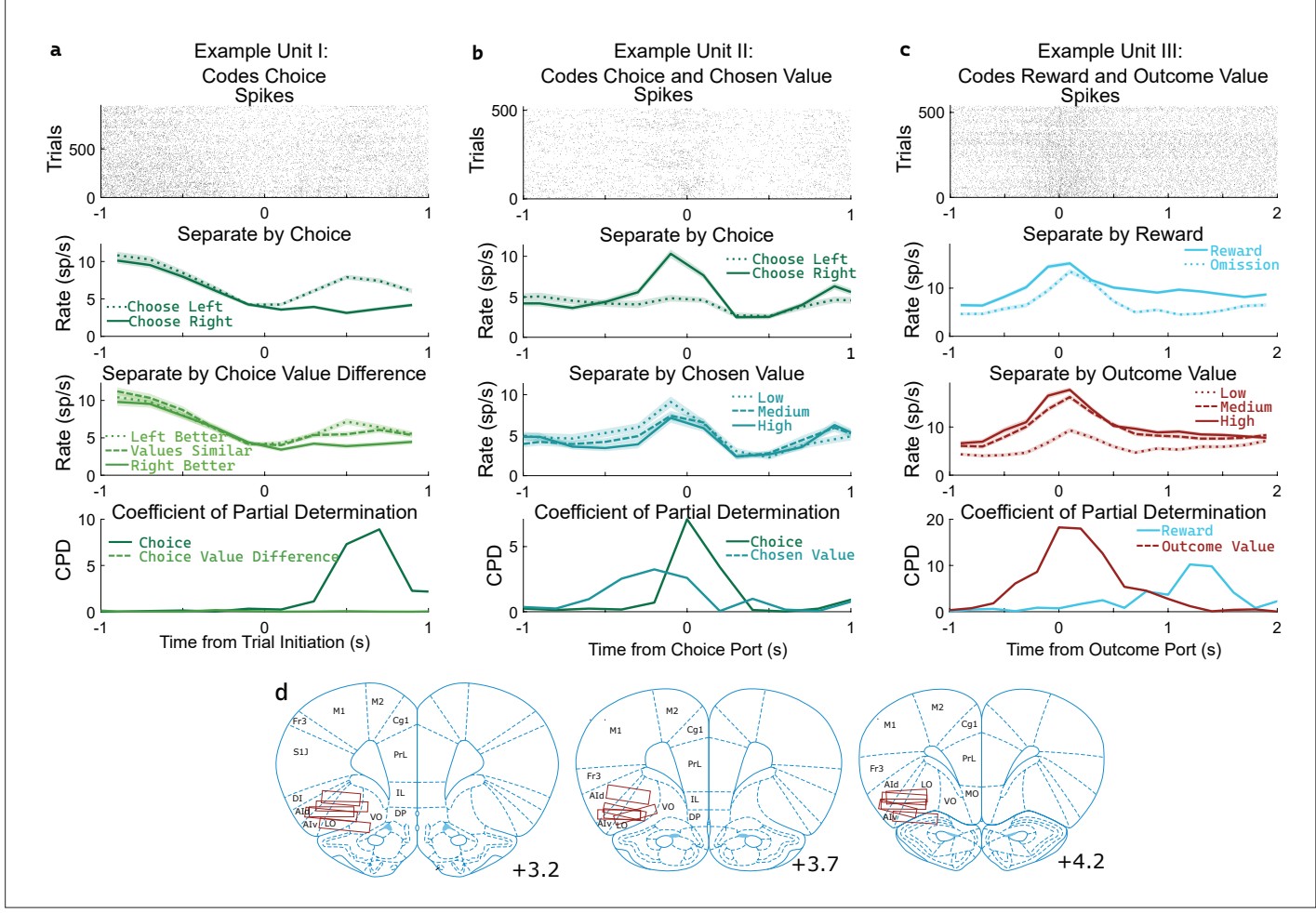

**Figure 3.** OFC units encode multiple correlated variables.

(**a**) Example unit whose firing rate differs both with the rat's choice and with the difference in value between the two possible choices. An analysis using coefficient of partial determination (CPD) reveals choice coding, but no coding of expected value. (**b**) Example unit whose firing rate differs both with the rat's choice and with the expected value of the choice port visited on that trial. CPD analysis reveals coding of both of these variables, though with different timecourses. (**c**) Example unit whose firing rate differs both with reward received and with the expected value of the outcome port visited. CPD analysis reveals coding of both of these variables, with different timecourses. (**d**) Approximate location of recording electrodes targeting OFC, which in rats is represented by regions LO and AIv (***Paxinos and Watson, 2006***; ***Price, 2007***; ***Stalnaker et al., 2015***), estimated using histology images (***Figure 3—figure supplement 1***).

The online version of this article includes the following figure supplement(s) for figure 3:

**Figure supplement 1.** Histological verification of implant locations in OFC.

**Figure supplement 2.** Correlations among predictors in the model used for analysis of electrophysiology data.

## Neural activity in orbitofrontal cortex codes expected value of outcomes

We implanted a recording array into the left OFC (***Figure 3d***, ***Figure 3—figure supplement 1***) of each of six rats, and performed electrophysiological recordings during 51 behavioral sessions. This dataset yielded 477 activity clusters, including both single-unit and multi-unit recordings. We characterized whether the activity of each unit encoded three different types of value information, which we estimated trial-by-trial using fits of the cognitive model (***Daw, 2011***). The first type was the difference in expected value between the left and the right choice ports, which we term 'choice value difference'. Representations of this kind would be consistent with a role in driving choice. The second was the expected value of the choice port that the rat actually selected on that trial, termed 'chosen value'. This signal has also been proposed to play a role in the choice process (***Rustichini and Padoa-Schioppa,***

*2015*). The last was the expected value of the outcome port visited by the rat on that trial, which we term 'outcome value'. Representations of outcome value are predictions about immediate reward probability, and are consistent with a role in learning. We sought to determine whether our neural recording data contained correlates of choice-related value signals, outcome-related value signals, or both.

Determining whether a unit encodes these quantities requires separating their influence from that of other variables with which they may be correlated. For example, if the firing rate of a unit differs between left-choice and right-choice trials, it will also differ between trials in which the left port had a higher value and those in which the right port had a higher value (*Figure 3a*). This happens because the rat is more likely to select the choice port that is currently higher-valued. Similarly, if the firing rate of a unit differs between rewarded and unrewarded trials, it will also differ between trials with a high and a low outcome port value (*Figure 3c*). To quantify coding strength in light of these and other correlations (*Figure 3—figure supplement 2*), we used a multiple regression approach. Specifically, we fit a separate regression model to predict the spike counts of each unit in each of several 200ms time bins taken relative to the port entry events, using as regressors the potentially learning-related events from one trial (choice port, outcome port, reward, and their interactions, as well as outcome value) as well as the potentially choosing-related variables from the next trial (choice port, choice value difference, and chosen value). We then computed for each regressor the coefficient of partial determination (CPD, also known as 'partial r-squared'; *Cai et al., 2011*; *Kennerley et al., 2011*), which quantifies the percentage of remaining variance explained by that regressor, once the influence of all other regressors has been accounted for (*Figure 3abc*, bottom row).

Coefficients of partial determination can be computed for a particular fit (one unit in one time bin), or for a collection of fits (aggregating variance over units, bins, or both). First, we considered coding in individual clusters (single- or multi-unit), aggregating data from time bins within a one-second window around the time of each nose port entry. We assess significance by constructing for each unit a set of null datasets by circularly permuting the trial labels, and comparing the CPDs in the true dataset to those in the null datasets. Using this method, we found that a large fraction of units significantly modulated their firing rate according to outcome-value, with the largest fraction doing so at outcome port entry (170/477, 36%; permutation test at p<0.01). In contrast, a relatively small fraction of units modulated their firing rate according to choice-value-difference (largest at outcome port entry: 34/477, 7%), or to chosen-value (largest at choice port entry: 61/477, 13%). Furthermore, the magnitude of CPD was larger for outcome-value than for the other value regressors. Considering the port entry event with the strongest coding for each regressor, the mean cluster had CPD for outcome-value 3.5 x larger than for choice-value-difference (p=$10^{-24}$, sign rank test; median unit 1.6 x larger; *Figure 4a*, note logarithmic axes), and 3.7 x larger than for chosen-value (p=$10^{-14}$; median unit 1.7 x; *Figure 4a*). Considering each 200ms time bin separately, we find the fraction of units encoding outcome value rises shortly before step two initiation and peaks sharply around the time of outcome port entry, the fraction coding choice value difference peaks at outcome port entry as well, while the fraction coding chosen value peaks at the time of choice port entry (*Figure 4—figure supplement 1*). Next, we considered coding at the population level, computing CPD over all clusters for each time bin and subtracting the average CPD from the null datasets (*Figure 4b*). We found robust population coding of outcome-value, beginning at the time of entry into the bottom-center port, and peaking shortly after entry into the outcome port at 0.82%. In contrast, population coding of choice-value-difference and chosen-value was low in all time bins, reaching a maximum of only 0.29%. Population coding in the OFC was also present for other regressors in our model, especially for reward, choice, and outcome port (*Figure 4c*).

Similar results were found when considering single- and multi-unit clusters separately (*Figure 4—figure supplement 2*, *Figure 4—figure supplement 3*), as well as when separating the units recorded from individual rats (*Figure 4—figure supplement 4*, *Figure 4—figure supplement 5*). These results were robust to replacing the choice-related value regressors with analogs that consider the full decision variable, including contributions from perseveration and novelty preference as well as expected value (*Figure 4—figure supplement 6*). They were also robust to removing a regressor (interaction of reward and outcome port) that was particularly highly correlated with choice value difference (*Figure 4—figure supplement 6*). Across all of these variants a clear pattern was present: neural activity in OFC encodes the expected value of the visited outcome port more strongly than it encodes

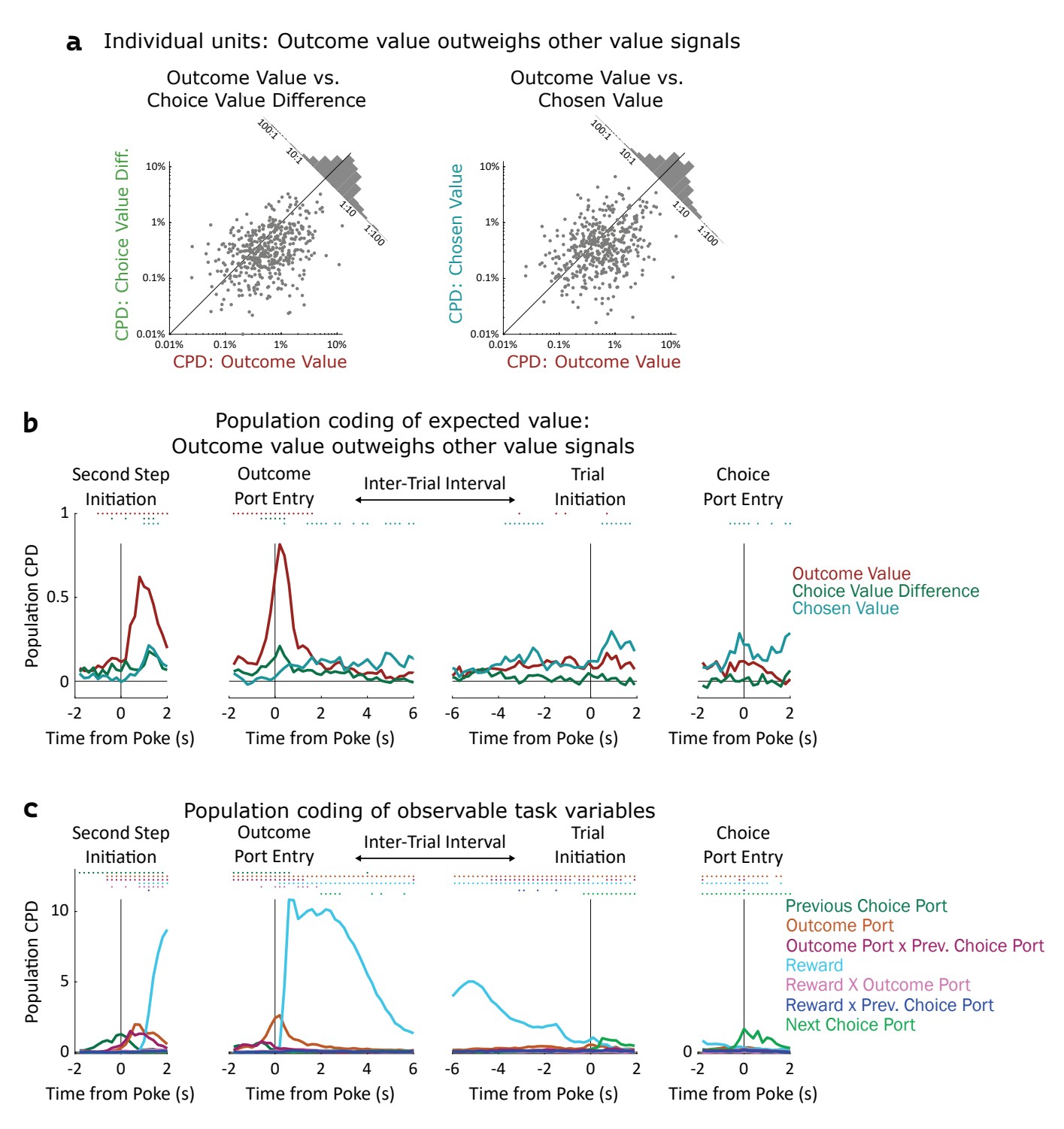

**Figure 4.** Coding of expected value of outcomes outweighs coding of expected value of choices. (**a**) Left: Scatterplot showing CPD for each unit (n=477) for the outcome-value regressor against CPD for the choice-value-difference regressor, both computed in a one-second window centered on entry into the outcome port. Right: Scatterplot showing CPD for outcome-value, computed at outcome port entry, against CPD for the chosen-value regressor, computed at choice port entry. (**b**). Timecourse of population CPD for the three expected value regressors. We have subtracted from each CPD the mean CPD found in permuted datasets. (**c**) Timecourse of population CPD for the remaining regressors in the model, which reflect observable variables and interactions between them.

The online version of this article includes the following figure supplement(s) for figure 4:

*Figure 4 continued on next page*

*Figure 4 continued*

**Figure supplement 1.** Fraction of units significantly encoding each regressor in each time bin.

**Figure supplement 2.** Coefficients of partial determination for value regressors, separately for single-unit and multi-unit clusters.

**Figure supplement 3.** Timecourse of population CPD for the regressors in our model, considering only single-unit clusters (above) or considering only multi-unit clusters (below).

**Figure supplement 4.** Analysis considering each rat individually.

**Figure supplement 5.** Fraction of significant units, considering each rat individually.

**Figure supplement 6.** Analysis removing the outcome-by-reward interaction regressor.

**Figure supplement 7.** Analysis using alternative choice-related value regressors.

**Figure supplement 8.** Correlates of reward prediction error.

either type of value information about the choice ports. In computational models (*Figure 2a*), this type of value information plays a role in supporting learning, but does not play a direct role in choice. Our neural recording results therefore suggest that the OFC may play a role in supporting learning, but cast doubt on the idea that they play a strong role in driving choice directly.

## Inactivations of OFC impair update process, not choice process

To assess the causal role of the OFC's value signals, we silenced neural activity using the optogenetic construct halorhodopsin (eNpHR3.0; *Figure 5b*, *Figure 5—figure supplement 1*) during either the *outcome period* (beginning at entry into the outcome port and lasting until the end of reward consumption), the *choice period* (beginning at the end of reward consumption and lasting until entry into the choice port on the subsequent trial), or *both periods* (*Figure 5a*, *Figure 5—figure supplement 2*) in an experimental group of nine rats (577 total sessions). Previous work (*Miller et al., 2017*) had shown that whole-session pharmacological silencing of the OFC specifically attenuates the 'planning index' (*Figure 2e*, see Methods), which quantifies the extent to which the rats' choices are modulated by past trials' outcomes in a way consistent with planning. Here, we found that optogenetic inactivation spanning both the outcome and the reward period reproduced this effect, decreasing the planning index on the subsequent trial (p=0.007, t-test with n=9 rats; *Figure 5c*). Inactivation during the outcome period alone also decreased the planning index on the subsequent trial (p=0.0007, *Figure 5c*), but inactivation during the choice period alone did not (p=0.64). Comparing the strength of the effect across time periods, we found that both reward-period and both-periods inactivation produced effects that were similar to one another (p=0.5), but greater than the effect of choice-period inactivation (p=0.007, p=0.02). We repeated the experiment in a control group of four rats (109 total sessions) that were implanted with optical fibers but did not express halorhodopsin. This sham inactivation produced no significant effect on the planning index for any time period (all p>0.15, t-test with n=4 rats; *Figure 5c*, grey diamonds), and experimental and control rats differed in the effects of outcome-period and both-period inactivation (p=0.02, p=0.02, two-sample t-tests). As in our previous study, inactivation of OFC did not significantly affect other regression-based behavioral indices (*Figure 5—figure supplement 3*). Together, these results indicate that silencing the OFC at the time of the outcome (and therefore the time of peak outcome-value coding, *Figure 4b*) is sufficient to disrupt planning behavior.

To help understand which aspect of the behavior was affected by silencing the OFC, we used our cognitive model (*Figure 2a*) to perform three different types of synthetic inactivation experiments. Each of these produced a distinct pattern of behavior, most clearly visible when we separately computed the contribution to the planning index of the past three trials' outcomes (*Figure 5e*). To simulate an effect of inactivation on the choice process, we decreased the value of the planning agent's weighting parameter ($\beta_q$; *Figure 2a*, 'choose a choice port') on synthetic inactivation trials, effectively reducing the influence of choice port value representations on choice. This resulted in choices that were more noisy in general, affecting the influence of all past outcomes (*Figure 5e*, left). To simulate effects of inactivation on the learning process, we reparameterized the agent's learning equation (*Figure 2a*, 'update outcome port values') to facilitate trial-by-trial alterations of learning parameters:

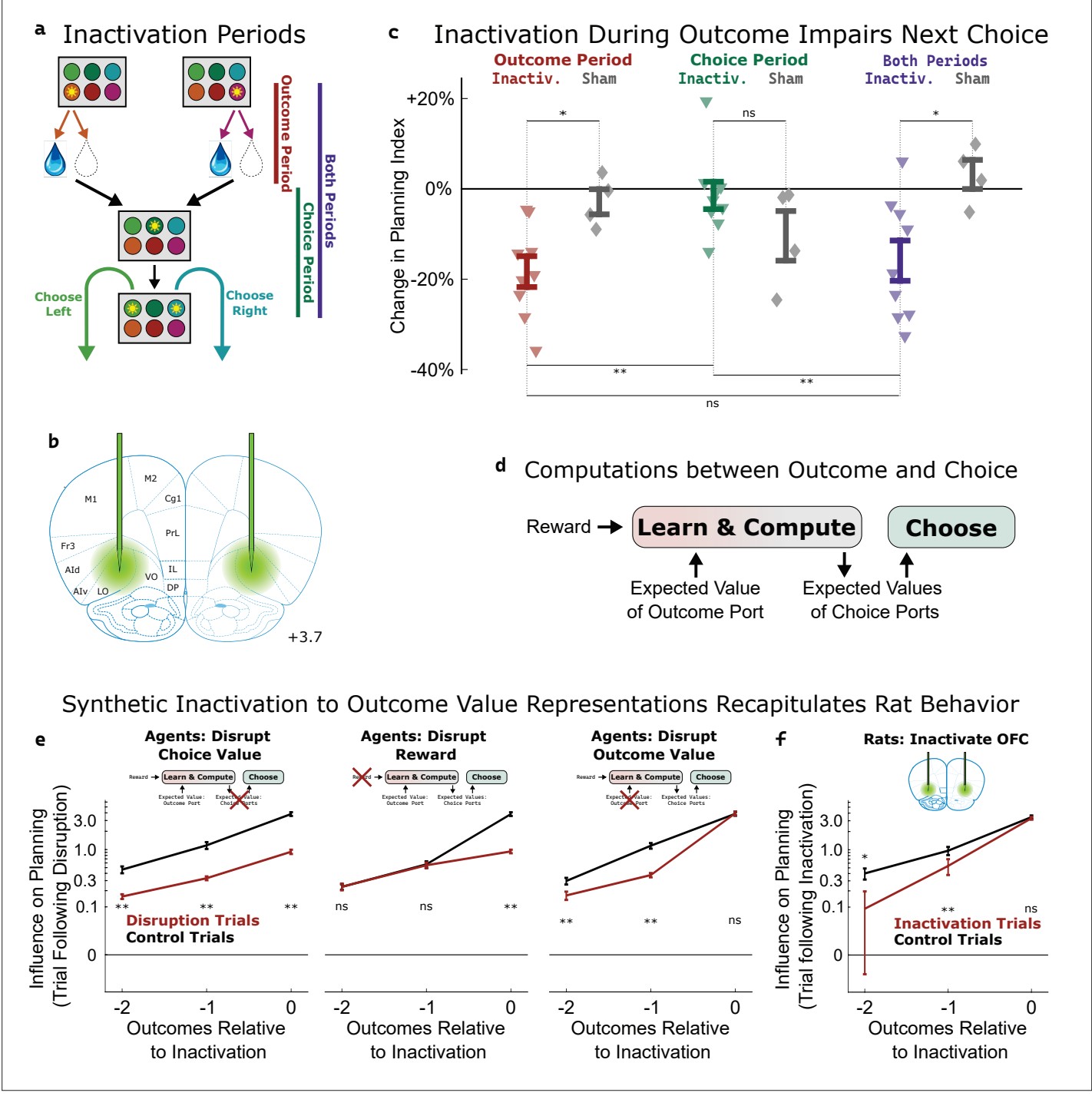

**Figure 5.** Inactivation of OFC attenuates influence of outcome values. (**a**) Three time periods of inactivation. Outcome-period inactivation began when the rat entered the outcome port, and continued until the rat exited the port, or for a minimum of two seconds. Choice-period inactivation began after this outcome period, and continued until the rat entered the choice port on the next trial, or for a maximum of 15 s. Both-period inactivation encompassed both of these periods. (**b**) Target location for optical fiber implants. See *Figure 5—figure supplement 1* for estimated actual locations in individual rats. Coronal section modified from *Paxinos and Watson, 2006*. (**c**) Effects of inactivation on the planning index on the subsequent trial for experimental rats (n=9, colored triangles) and sham-inactivation rats (n=4, gray diamonds). Bars indicate standard errors across rats. (**d**) Simplified schematic of the representations and computations that take place in our software agent between the delivery of the outcome on one trial and the choice on the next. Compare to *Figure 2a*. (**e**) Analysis of synthetic datasets created by disrupting different representations within the software agent on a subset of trials. Each panel shows the contribution to the planning index of trial outcomes at different lags on choices, both on control trials (black)

*Figure 5 continued on next page*

*Figure 5 continued*

and on trials following disruption of a representation (red). Bars indicate standard error across simulated rats (see Methods) (**f**) Same analysis as in c, applied to data from optogenetic inactivation of the OFC during the outcome period.

The online version of this article includes the following figure supplement(s) for figure 5:

**Figure supplement 1.** Locations of optogenetics Implants.

**Figure supplement 2.** Periods of optogenetic inhibition.

**Figure supplement 3.** Effects of optogenetics on all behavioral indices.

**Figure supplement 4.** Average fit weights of the trial-history regression model to behavioral data from the optogenetics experiment.

**Figure supplement 5.** Fit weights of the trial history regression model for individual rats in the optogenetics experiment.

$$V(O_t) \leftarrow \alpha_{value} V(O_t) + \alpha_{reward} - R_t + (1 - \alpha_{value} - \alpha_{reward}) - E[V]$$

where $\alpha_{value}$ and $\alpha_{reward}$ are now separate learning rate parameters, and E[$V$] represents the expected reward of a random-choice policy. We made a corresponding change to the symmetric update for the outcome port that was not visited (see Methods). To simulate an effect of inactivation on the role of reward in learning, we decreased $\alpha_{reward}$ on synthetic inactivation trials. This resulted in a change specifically to the influence of the outcome during which we performed the synthetic inactivation on future choice (*Figure 5e*, middle). To simulate an effect of inactivation on the role of value in learning, we decreased $\alpha_{value}$ on synthetic inactivation trials. This resulted in a change to the influence of outcomes from *previous* trials (*Figure 5e* right) but not of the trial during which synthetic inactivation was actually performed. This is because value acts as a summarized memory of previous trials' outcomes, and attenuating it affects the influence of all of these.

We performed the same analysis on the inactivation data from our rats (*Figure 5f*), and found that silencing OFC during the outcome period on a particular trial did not affect the influence of that trial's outcome on the upcoming choice (p=0.2, paired t-test with n=9 rats), but that it did affect the influence of the previous two trials' outcomes (p=0.004, p=0.02). This pattern was consistent with the synthetic dataset in which outcome value representations had been attenuated, but not the other synthetic datasets (compare *Figure 5f* to *Figure 5e*). We conclude that silencing the OFC in our task predominantly impairs the use of outcome port value information, needed for updating value expectations, but has little or no effect on the use of choice port value information, needed for driving choice.

## Discussion

In formal models of value-based cognition, value representations can play two very different computational roles. First, they can drive choosing, as expected values of different available options are compared to one another, and the best is selected. Secondly, they can drive learning, as the expected value of an outcome is compared to the reward actually received, and future expectations are updated. Value representations in the OFC have been reported in many tasks and species, but it is still unclear whether they drive one process, the other process, or both. The rat two-step task gave us the opportunity to separate the two roles, both in terms of coding in neural activity and in terms of the behavioral impact of silencing that activity. Using this approach, we find weak (though significant) representation of values associated with the available choices ('choice values'), little or no effect of silencing OFC at the putative time of choice, and effects of silencing inconsistent with impairing choice values in a computational model. Instead, we find strong representation of values associated with immediately impending reward outcomes ('outcome values'), a strong behavioral effect of silencing OFC at the time of those representations, and effects of silencing that are consistent with specifically impairing the use of outcome values for learning. This pattern of results suggests a much stronger role for OFC in learning than in choosing. More specifically, the results suggest that value information in the rodent OFC may not drive choice directly, but instead provide information about expected outcomes to a separate learning process. In the computational model most consistent with our data, OFC inactivation does not prevent learning, but instead impairs one of the critical inputs to the learning process, namely expected outcome value. Our results therefore are consistent with the idea that learning does not take place in the OFC itself, but in some other structure to which OFC supplies this key input.

These results clarify the role of a neural signal that has been reported over a wide variety of tasks and species (*O'Doherty, 2007*; *Wallis, 2012*; *Rudebeck and Murray, 2014*; *Stalnaker et al., 2015*). In decision tasks, several distinct types of value representation have been reported, but the most commonly observed is 'chosen value' (*Wallis and Miller, 2003*; *Sul et al., 2010*; *Kennerley et al., 2011*): the expected reward associated with the option that the subject chose. The role of this signal is not clear, it has been interpreted both as representing a post-decision signal that provides input to a learning process (*Schoenbaum et al., 2011*), but also as an internal component of the choice process itself (*Rustichini and Padoa-Schioppa, 2015*). These roles are difficult to fully disentangle using classic tasks, in which both learning and choosing operate over the same set of items. Rat behavior in the two-step task separates these possibilities. If 'chosen value' relates to the choice process, we would expect to see representations of the expected value of the port that was chosen (not learned about), while if it relates to the learning process, we would expect to see expected value of the port that was learned about (but not chosen). Our data predominantly provide evidence for the latter alternative in rats.

Our results build on prior work attempting to separate learning and choosing in trial-by-trial tasks in primates. One set of studies used a probabilistic reward-learning task (the 'three-armed bandit' task), along with detailed trial-by-trial analysis to quantify deficits in 'credit assignment', attributable to an impaired learning process, as well as deficits in decision-making (*Walton et al., 2010*). Studies using this tool reported that both macaques and humans with lesions to the OFC showed impairments in credit assignment, while those with lesions to a neighboring region, ventromedial prefrontal cortex (vmPFC), showed impairments in decision-making (*Noonan et al., 2010*; *Noonan et al., 2017*). A methodological limitation of these studies is that they employed nonselective lesions, which damage both local neurons (from which electrophysiological signals are recorded) as well as fiber tracts connecting other brain regions (from which they typically are not). A subsequent study *Rudebeck et al., 2017* used excitotoxic lesions, which do not damage passing fibers, and found that lesions spanning both OFC and vmPFC affected neither credit assignment nor decision making on this task. Instead, the study found that lesions of another neighboring region, ventrolateral prefrontal cortex (vlPFC), caused impairments in credit assignment (*Rudebeck et al., 2017*). Together, these primate results suggest that neural activity in OFC itself is not necessary either for normal choosing or for normal learning on the three-armed bandit task, but that vlPFC may play a specific role in supporting learning (*Murray and Rudebeck, 2018*). Mapping these findings to the rodent brain is not entirely straightforward. While the distinction in rodents between OFC and vmPFC is relatively clear, the distinction between OFC and vlPFC is much less so (*Price, 2007*). Anatomically, many of the criteria that motivate the proposed homology between rodent OFC (typically taken to mean areas LO and AIv) and primate OFC (typically taken to mean areas 11 l and 13) may apply to parts of primate vlPFC (especially area 12) as well (*Price, 2007*; *Stalnaker et al., 2015*). Like in OFC, neurons in vlPFC have been shown to represent expected outcomes (*Kobayashi et al., 2010*; *Rich and Wallis, 2014*), though these signals have not been characterized as intensively as those in primate OFC. A possibility is that primate OFC and vlPFC are specialized for functions that in rodents are both performed by the OFC.

Our results also build on those of previous studies that attempt to separate learning and choosing by separating these functions in time, typically by tens of minutes or longer. Here, however, we investigated a very different timescale in that learning and choosing were interleaved, both occurring on each trial of the two-step task. Our inactivation results are consistent with previous studies suggesting a role for the OFC in learning (*Takahashi et al., 2009*; *Jones et al., 2012*), but in apparent tension with others finding that inactivation impairs performance on a post-learning assay (*Jones et al., 2012*; *Gremel and Costa, 2013*; *Murray et al., 2015*). One possibility is that the OFC plays different roles for different timescales, or for these different types of behavior. Another possibility is that the OFC plays a common role across timescales and tasks, supporting a process that updates choice mechanisms – in long-timescale tasks this process may take place during either the learning or the probe session, while in our task it necessarily takes place between the outcome of one trial and the choice on the next.

Our results are consistent with the view that the OFC carries expectancy signals (*Schoenbaum and Roesch, 2005*; *Rudebeck and Murray, 2014*) indicating which outcomes are expected to follow from the current state. This view is distinct from the 'chosen value' view described earlier in that it claims

that these signals indicate the particular identities of the expected outcomes (e.g. different types of foods) rather than the abstract 'reward' or 'common currency' value found in reinforcement learning or neuroeconomic theories. Our experiment does not speak to the difference between these views, because the only reward available in our task is a water droplet of a fixed size. Our 'outcome value' correlates might therefore reflect expectations of this reward (the water droplet) in particular, and play a role in updates based on this expectation. In formal models, this might be described as updating estimates of a state transition probability (e.g. from an 'at the outcome port' state to a 'receive water droplet' state). Alternatively, 'outcome value' might abstract over many different possible rewarding outcomes. In formal models, this might be described as updating estimates of the reward function (e.g. in an 'at the outcome port' state).

Our results are also in tension with those of some studies using economic choice tasks, in which subjects make decisions between pairs of well-learned stimuli leading to different quantities of differently flavored rewards. A pair of studies, one in mice (*Kuwabara et al., 2020*) and one in primates (*Ballesta et al., 2020*), uses methods that disrupt OFC neural activity on a particular subset of trials, and reports that subjects' decisions are disrupted specifically on those trials. Recording studies in these tasks, both in primates and in mice (*Padoa-Schioppa and Assad, 2006*; *Kuwabara et al., 2020*), have reported neural correlates of the values of the individual options that are available. This 'offer value' correlate is distinct from the 'chosen value' correlate discussed earlier, and is suitable for a role in driving decisions directly. These results have led to the view that OFC value representations, at least in these tasks and species, play a key role in decision-making (*Rustichini and Padoa-Schioppa, 2015*; *Padoa-Schioppa and Conen, 2017*). Our results indicate that in the rat two-step task, OFC does not play this role, since choice-related value correlates are minimal, and inactivation effects are inconsistent with a direct role in choice. One possible resolution to this tension is a difference between tasks. It has been proposed that OFC plays a role specifically in choices between outcomes that offer different 'goods' (*Padoa-Schioppa, 2011*; e.g. different flavors *Kuwabara et al., 2020*). In primate tasks involving decisions between outcomes that differ in other properties – such as magnitude, probability, or delay – OFC value representations are typically more consistent with chosen-value than with offer-value coding (*Wallis and Miller, 2003*; *Roesch and Olson, 2004*; *Rich and Wallis, 2016*). Casting doubt on this goods-based interpretation are a set of recent results from a flavor-based task in rats (*Gardner et al., 2017*; *Gardner et al., 2020*) showing that silencing the OFC does not affect choice. Another possible resolution is a difference between species. Consistent with either of these possibilities are recent results from a choice task in rats in which options differed in probability and magnitude (but not flavor): rat OFC in this task did not carry correlates of offer value, and silencing impaired trial-history effects rather than the influence of offers on choice (*Constantinople et al., 2019*).

A final possibility is that OFC contributes to a model-based 'policy update' process that combines known information about the structure of the world with new information, in order to modify an animal's behavioral strategy (*Gershman et al., 2014*; *Ludvig et al., 2017*; *Mattar and Daw, 2018*). Such a process could happen at different times in different tasks. In our task, policy update might happen between the end of one trial and the beginning of the next, and be interpreted as learning. In economic choice tasks, policy update might happen immediately after the options are presented, but before a choice is made, and be interpreted as related to deliberation. Indeed, experiments analyzing response times suggest that such a deliberative process sometimes occurs during economic choice (*Krajbich et al., 2010*), and may be related to moment-by-moment patterns of activity in the OFC (*Rich and Wallis, 2016*).

Consistent with this idea are results from a series of recent studies using a flavor-based economic choice task for rats (*Gardner et al., 2017*). If rats are pre-fed rewards of one flavor immediately before a task session begins, they normally alter their choice preferences during that session; silencing OFC activity impairs this update (*Gardner et al., 2019*). If rats are presented with a novel pairing of flavors that have previously been experienced separately, their preferences are initially unstable, but normally converge over the course of a session; silencing OFC activity impairs this update as well (*Gardner et al., 2020*). These results indicate that the OFC activity is crucial specifically at times when a behavioral strategy is being updated. They leave open the question of what computational role this activity plays. They are consistent with several possible roles, including identifying when the strategy should be updated, driving the update of that strategy directly, or driving decisions between recently updated alternatives. Our analysis of the coarse effects of

inactivation in different time windows replicates this result. Our analysis of the fine-grained effects of inactivation builds on it, using a cognitive model to jointly interpret neural recordings and trial-by-trial behavioral effects of inactivation. They indicate that the detailed effects of inactivation are attributable in particular to degradation of the outcome value signal (and not, for example, the reward signal, which is stronger and carried by a larger fraction of clusters). This suggests that, at least in our task, the OFC plays a particular computational role in supporting update: it provides outcome value estimates to a learning system, elsewhere in the brain, that updates the behavioral policy.

Our data are also broadly consistent with the idea that the OFC represents a 'cognitive map' of task-relevant state information. A broad version of this account (*Wilson et al., 2014*) was based largely on lesion and inactivation data, and leaves open the question of whether this cognitive mapping information is used to drive choice directly, to guide learning, or for both. A narrower version of this cognitive mapping account was based also on fMRI data in human subjects (*Chan et al., 2016*; *Schuck et al., 2016*), and proposes that activity in the OFC specifically represents the subject's beliefs about the current unobservable state of the environment. In our task, the most relevant states are fully observable (indicated by which noseports are currently lit). However it is possible to view the current reward probability contingencies (right/left outcome ports rewarded at 80%/20% vs. 20%/80%) as an unobservable task-relevant state variable that the animal might make inferences about. In our model, the 'choice value difference' variable corresponds very closely to the current belief about this unobservable state. The narrow version of the cognitive mapping account might therefore predict that we would see this variable strongly represented in OFC throughout the entire trial. Our finding that it is represented only transiently and weakly puts our data in apparent tension with the narrower cognitive mapping account. A possible resolution to this tension lies in a difference in brain regions. The fMRI data (*Chan et al., 2016*; *Schuck et al., 2016*) find their strongest representations of unobservable state in very medial regions of the orbital surface and in regions of the medial surface, likely belonging to the medial network (areas 10 and 11 m; *Price, 2007*) rather than in the orbital network regions (areas 11 l, 12/47, and 13) which are most plausibly homologous to the regions we investigated here (LO and AIv). A promising direction for future work would be to perform experiments analogous to ours in the regions of the rodent brain that are plausibly homologous to ventral parts of the medial network (VO and MO) as well as elsewhere in PFC.

In summary, we find that rat behavior on a multi-step decision task dissociates the computational role of expected value information in choosing from its role in learning. We investigate the role of expected value representations in the OFC, and find a pattern of results indicating that these representations do not directly drive choice in this task, but instead support a learning process that updates choice mechanisms elsewhere in the brain. In addition to clarifying the function of the rodent OFC, these results provide insight into the overall architecture of value-based cognition. Specifically, they reveal that it is modular: that value-based learning and value-based choosing depend on separate neural representations. This lends credence to computational models which separate these functions (*Joel et al., 2002*; *Song et al., 2017*). It also motivates a search for the neural modules in the rat which play the remaining computational roles, including the role of directly driving choice.

## Methods
### Subjects

All subjects were adult male Long-Evans rats (Taconic Biosciences; Hilltop Lab Animals), placed on a restricted water schedule to motivate them to work for water rewards. Rats were housed on a reverse 12 hr light cycle and trained during the dark phase of the cycle. Rats were pair housed during behavioral training and then single housed after being implanted with microwire arrays or optical fiber implants. All experimental procedures were performed in strict accordance with the recommendations in the Guide for the Care and Use of Laboratory Animals of the National Institutes of Health., and were approved by the Princeton University Institutional Animal Care and Use Committee (protocol #1853). No explicit power analysis was used to determine the number of rats; instead we aimed for a sample size consistent with previous studies using similar tools.

## Two-step behavioral task

Rats were trained on a two-step behavioral task, following a shaping procedure which has been previously described (*Miller et al., 2017*). Rats performed the task in custom behavioral chambers containing six 'nose ports' arranged in two rows of three, each outfitted with a white LED for delivering visual stimuli, as well as an infrared LED and phototransistor for detecting rats' nose entries into the port. The left and right ports in the bottom row also contained sipper tubes for delivering water rewards. The rat initiated each trial by entering the illuminated top center port, causing the two top side ports ('choice ports') to illuminate. The rat then made his choice by entering one of these ports. Immediately upon entry into a choice port, two things happened: the bottom center port light illuminateed, and one of two possible sounds began to play, indicating which of the two bottom side ports ('outcome ports') would eventually be illuminated. The rat then entered the bottom center port, which caused the appropriate outcome port to illuminate. Finally, the rat entered the outcome port which illuminated, and received either a water reward or an omission. Once the rat had consumed the reward, a trial-end sound played, and the top center port illuminated again to indicate that the next trial was ready.

The selection of each choice port led to one of the outcome ports becoming available with 80% probability (common transition), and to the other becoming available with 20% probability (uncommon transition). These probabilities were counterbalanced across rats, but kept fixed for each rat for the entirety of his experience with the task. The probability that entry into each bottom side port would result in reward switched in blocks. In each block one port resulted in reward 80% of the time, and the other port resulted in reward 20% of the time. Block shifts happened unpredictably, with a minimum block length of 10 trials and a 2% probability of block change on each subsequent trial.

## Analysis of behavioral data: Planning index and model-free index

We quantify the effect of past trials and their outcomes on future decisions using a logistic regression analysis based on previous trials and their outcomes (*Lau and Glimcher, 2005*; *Miller et al., 2017*). We define vectors for each of the four possible trial outcomes: common-reward (CR), common-omission (CO), uncommon-reward (UR), and uncommon-omission (UO), each taking on a value of +1 for trials of their type where the rat selected the left choice port, a value of –1 for trials of their type where the rat selected the right choice port, and a value of 0 for trials of other types. We use the following regression model:

$$
\begin{aligned}
log\left(\frac{P_{left}(t)}{P_{right}(t)}\right) = \sum_{\tau=1}^{T} \beta_{CR}(\tau) \cdot CR(t-\tau) + \sum_{\tau=1}^{T} \beta_{CO}(\tau) \cdot CO(t-\tau) + \\
\sum_{\tau=1}^{T} \beta_{UR}(\tau) \cdot UR(t-\tau) + \sum_{\tau=1}^{T} \beta_{UO}(\tau) \cdot UO(t-\tau)
\end{aligned}
\tag{1}
$$

where $\beta_{cr}$, $\beta_{co}$, $\beta_{ur}$, and $\beta_{uo}$ are vectors of regression weights which quantify the tendency to repeat on the next trial a choice that was made trials ago and resulted in the outcome of their type, and $T$ is a hyperparameter governing the number of past trials used by the model to predict upcoming choice, which was set to 3 for all analyses.

We expect model-free agents to repeat choices which lead to reward and switch away from those which lead to omissions *Daw et al., 2011*, so we define a model-free index for a dataset as the sum of the appropriate weights from a regression model fit to that dataset:

$$
ModelFreeIndex = \sum_{\tau=1}^{T} [\beta_{CR}(\tau) + \beta_{UR}(\tau)] - \sum_{\tau=1}^{T} [\beta_{UO}(\tau) + \beta_{CO}(\tau)]
\tag{2}
$$

We expect that planning agents will show the opposite pattern after uncommon transition trials, since the uncommon transition from one choice is the common transition from the other choice. We define a planning index:

$$
PlanningIndex = \sum_{\tau=1}^{T} [\beta_{CR}(\tau) - \beta_{UR}(\tau)] + \sum_{\tau=1}^{T} [\beta_{UO}(\tau) - \beta_{CO}(\tau)]
\tag{3}
$$

We also compute the main effect of past choices on future choice

$$StayIndex = \sum_{\tau=1}^{T} [\beta_{CR}(\tau) + \beta_{UR}(\tau) + \beta_{UO}(\tau) + \beta_{CO}(\tau)] \qquad (4)$$

As well as an index quantifying the tendency to repeat choices the lead to common transitions and to switch away from those that lead to uncommon transitions:

$$Common - Stay/Uncommon - Switch = \sum_{\tau=1}^{T} [\beta_{CR}(\tau) - \beta_{CO}(\tau)] + \sum_{\tau=1}^{T} [\beta_{UR}(\tau) - \beta_{UO}(\tau)] \qquad (5)$$

## Mixture-of-agents behavior model

We model behavior and obtain trial-by-trial estimates of value signals using an agent-based computational model similar to one that we have previously shown to provide a good explanation of rat behavior on the two-step task (*Miller et al., 2017*). This model adopts the mixture-of-agents approach, in which each rat's behavior is described as resulting from the influence of a weighted average of several different 'agents' implementing different behavioral strategies to solve the task. On each trial, each agent $A$ computes a value, $Q_A(c)$, for each of the two available choices $c$, and the combined model makes a decision according to a weighted average of the various strategies' values, $Q_{total}(c)$:

$$Q_{total(c)} = \sum_{A \in \{agents\}} \beta_A Q_A(c)$$
$$\pi(c) = \frac{e^{Q_{total(c)}}}{\sum_{c'} e^{Q_{total(c')}}} \qquad (6)$$

where the β's are weighting parameters determining the influence of each agent, and $(c)$ is the probability that the mixture-of-agents will select choice $c$ on that trial. The model which we have previously shown to provide the best explanation of rat's behavior contains four such agents: model-based temporal difference learning, novelty preference, perseveration, and bias. The model used here is identical to the one in our previous paper (*Miller et al., 2017*), except that the perseverative agent is modified to allow it to consider many past trials rather than only the immediately previous trial (*Miller et al., 2019*).

## Model-based temporal difference learning

Model-based temporal difference learning is a planning strategy, which maintains separate estimates of the probability with which each action (selecting the left or the right choice port) will lead to each outcome (the left or the right outcome port becoming available), $P(o|a)$, as well as the probability, $R(o)$, with which each outcome will lead to reward. This strategy assigns values to the actions by combining these probabilities to compute the expected probability with which selection of each action will ultimately lead to reward:

$$Q_{plan}(c) = \sum_{o} V(o)P(o|c) \qquad (7)$$

At the beginning of each session, the reward estimate $V(o)$ is initialized to 0.5 for both outcomes, and the transition estimate $P(o|c)$ is set to the true transition function for the rat being modeled (0.8 for common and 0.2 for uncommon transitions). After each trial, the reward estimate for both outcomes is updated according to

$$V(o) \leftarrow \begin{cases} (1 - \alpha_{plan}) \cdot V(o) + \alpha_{plan} \cdot r_t, \text{for } o = o_t \\ (1 - \alpha_{plan}) \cdot V(o) - \alpha_{plan} \cdot r_t, \text{for } o \neq o_t \end{cases} \qquad (8)$$

where $o_t$ is the outcome that was observed on that trial, $r_t$ is a binary variable indicating reward delivery, and α is a learning rate parameter constrained to lie between zero and one.

## Novelty preference

The novelty preference agent follows an 'uncommon-stay/common switch' pattern, which tends to repeat choices when they lead to uncommon transitions on the previous trial, and to switch away from them when they lead to common transitions. Note that some rats have positive values of the $\beta_{np}$ parameter weighting this agent (novelty preferring) while others have negative values (novelty averse; see *Figure 1e*):

$$
Q_{np}(ct) \leftarrow \begin{cases} 0, & \text{common transition trails} \\ 1, & \text{uncommon transition trials} \end{cases}
$$
$$
Q_{np}(c \neq c_t) \leftarrow 1 - Q_{np}(c_t) \tag{9}
$$

### Perseveration

Perseveration is a pattern which tends to repeat choices that have been made in the recent past, regardless of whether they led to a common or an uncommon transition, and regardless of whether or not they led to reward.

$$
Q_{persev}(c_t) < -(1 - \alpha_{persev})Q_{persev}(c_t) + \alpha_{persev}
$$
$$
Q_{persev}(c \neq c_t) < -(1 - \alpha_{persev})Q_{persev}(c \neq c_t) \tag{10}
$$

### Bias

Bias is a pattern which tends to select the same choice port on every trial. Its value function is therefore static, with the extent and direction of the bias being governed by the magnitude and sign of this strategy's weighting parameter $\beta_{bias}$.

$$
Q_{bias}(left) = 1
$$
$$
Q_{bias}(right) = -1 \tag{11}
$$

## Model fitting

We implemented the model described above using the probabilistic programming language Stan (*Carpenter, 2016*; *Lau, 2017*), and performed maximum-a-posteriori fits using weakly informative priors on all parameters (*Gelman et al., 2013*) The prior over the weighting parameters $\beta$ was normal with mean 0 and sd 0.5, and the prior over $\alpha$ was a beta distribution with $a=b = 3$. For ease of comparison, we normalize the weighting parameters $\beta_{plan}$, $\beta_{np}$, and $\beta_{persev}$, dividing each by the standard deviation of its agent's associated values ($Q_{plan}$, $Q_{np}$, and $Q_{persev}$) taken across trials. Since each weighting parameter affects behavior only by scaling the value output by its agent, this technique brings the weights into a common scale and facilitates interpretation of their relative magnitudes, analogous to the use of standardized coefficients in regression models.

## Surgery: Microwire array implants

Six rats were implanted with microwire arrays (Tucker-David Technologies) targeting OFC unilaterally. Arrays contained tungsten microwires 4.5 mm long and 50 μm in diameter, cut at a 60° angle at the tips. Wires were arranged in four rows of eight, with spacing 250 μm within-row and 375 μm between rows, for a total of 32 wires in a 1.125 mm by 1.75 mm rectangle. Target coordinates for the implant with respect to bregma were 3.1–4.2 mm anterior, 2.4–4.2 mm lateral, and 5.2 mm ventral (~4.2 mm ventral to brain surface at the posterior-middle of the array).

In order to expose enough of the skull for a craniotomy in this location, the jaw muscle was carefully resected from the lateral skull ridge in the area near the target coordinates. Dimpling of the brain surface was minimized following procedures described in more detail elsewhere (*Akrami et al., 2017*). Briefly, a bolus of petroleum jelly (Puralube, Dechra Veterinary Products) was placed in the center of the craniotomy to protect it, while cyanoacrylate glue (Vetbond, 3 M) was used to adhere the pia mater to the skull at the periphery. The petroleum jelly was then removed, and the microwire

array inserted slowly into the brain. Rats recovered for a minimum of 1 week, with ad lib access to food and water, before returning to training.

## Electrophysiological recordings

Once rats had recovered from surgery, recording sessions were performed in a behavioral chamber outfitted with a 32 channel recording system (Neuralynx). Spiking data was acquired using a bandpass filter between 600 and 6000 Hz and a spike detection threshold of 30 μV. Clusters were manually cut (Spikesort 3D, Neuralynx), and both single- and multi-units were considered. All manually cut units were used for analysis. We observed that a small fraction of trials showed apparent artifacts in which some units appeared to have extremely high firing rates, which we suspect was due to motion of the implant or tether. We therefore excluded units from analysis on trials where the Median Absolute Deviation (*Leys et al., 2013*) of their spike count exceeded a conservative threshold of three.

## Analysis of electrophysiology data

To determine the extent to which different variables were encoded in the neural signal, we fit a series of regression models to our spiking data. Models were fit to the spike counts emitted by each unit in 200ms time bins taken relative to the four noseport entry events that made up each trial. There were ten total regressors, defined relative to a pair of adjacent trials. Seven of them were binary (coded as +–1), and related to observable task variables: the choice port selected on the earlier trial(left or right), the outcome port visited (left or right), the reward received (reward or omission), the interaction between choice port and outcome port (common or uncommon transition), the interaction between choice port and reward, the interaction between outcome port and reward, and the choice port selected on the later trial. Three additional regressors were continuous and related to subjective reward expectation: the expected value of the outcome port visited (*V*) for the first trial, the value difference between the choice ports (*Q(left) - Q(right)*), and the value of the choice port selected (*Q(chosen)*) on the subsequent trial. These last three regressors were obtained using the agent-based computational model described above, with parameters fit separately to each rat's behavioral data. Regressors were z-scored to facilitate comparison of fit regression weights. Models were fit using the Matlab function glmnet *Qian, 2013* using a Poisson noise model. We fit each model both with no regularization and with L1 regularization ($\alpha$=1, $\lambda$ =0, $10^{-10}$, $10^{-9}$, …,$10^{-1}$), using the model with the weakest regularization that still allowed all weights to be identifiable.

In our task, many of these regressors were correlated with one another (*Figure 3—figure supplement 2*), so we quantify encoding using the coefficient of partial determination (CPD; also known as partial r-squared) associated with each (*Cai et al., 2011*; *Kennerley et al., 2011*). This measure quantifies the fraction of variance explained by each regressor, once the variance explained by all other regressors has been taken account of:

$$CPD(X_i, u, t) = \frac{(SSE(X_{-i}, u, t) - SSE(X_{all}, u, t))}{SSE(X_{-i}, u, t)} \quad (12)$$

where *u* refers to a particular unit, *t* refers to a particular time bin, and SSE($X_{all}$) refers to the sum-squared-error of a regression model considering all eight regressors described above, and SSE($X_{-i}$) refers to the sum-squared-error of a model considering the seven regressors other than $X_i$. We compute total CPD for each unit by summing the SSE associated with the regression models for that unit for all time bins:

$$CPD(X_i, u) = \frac{\sum_t (SSE(X_{-i}, u, t) - SSE(X_{all}, u, t))}{\sum_t SSE(X_{-i}, u, t)} \quad (13)$$

We report this measure for individual example units showing all time bins (*Figure 2*, bottom). We report the CPD for each unit for particular port entry events (*Figure 3a*), taking the sum over the five bins making up a 1 s time window centered on a particular port entry event (top neutral center port, choice port, bottom neutral center port, or outcome port). We report the 'population CPD' (*Figure 3b*, *Figure 3c*) by aggregating over all units for a particular time bin:

$$CPD(X_i, t) = \frac{\sum_u (SSE(X_{-i}, u, t) - SSE(X_{all}, u, t))}{\sum_u SSE(X_{-i}, u, t)} \quad (14)$$

For each of these measures, we assess significance by comparing the CPD computed on the true dataset to a distribution of CPDs computed on surrogate datasets constructed by circularly permuting the trial labels within each session. We use permuted, rather than shuffled, labels in order to preserve trial-by-trial correlational structure. If the true CPD is larger than most of the CPDs from these surrogate datasets, we can reject the null hypothesis that the CPD is driven by correlational structure alone. We compute a permutation p-value by finding the percentile of the true CPD within the distribution of CPDs in surrogate datasets. In the main-text figure (*Figure 3b*, *Figure 3c*) we additionally subtract the mean of the CPDs from the surrogate datasets, in order to give a measure that can be fairly compared to zero.

## Surgery: Optical fiber implant and virus injection

Rats were implanted with sharpened fiber optics and received virus injections following procedures similar to those described previously (*Hanks et al., 2015*; *Kopec et al., 2015*; *Akrami et al., 2017*), and documented in detail on the Brody lab website. A 50/125 μm LC-LC duplex fiber cable (Fiber Cables) was dissected to produce four blunt fiber segments with LC connectors. These segments were then sharpened by immersing them in hydroflouric acid and slowly retracting them using a custom-built motorized jig attached to a micromanipulator (Narashige International) holding the fiber. Each rat was implanted with two sharpened fibers, in order to target OFC bilaterally. Target coordinates with respect to bregma were 3.5 mm anterior, 2.5 mm lateral, 5 mm ventral. Fibers were angled 10 degrees laterally, to make space for female-female LC connectors which were attached to each and included as part of the implant.

Four rats were implanted with sharpened optical fibers only, but received no injection of virus. These rats served as uninfected controls.

Nine additional rats received both fiber implants as well as injections of a virus (AAV5-CaMKIIα-eNpHR3.0-eYFP; UNC Vector Core) into the OFC to drive expression of the light-activated inhibitory opsin eNpHR3.0. Virus was loaded into a glass micropipette mounted into a Nanoject III (Drummond Scientific), which was used for injections. Injections involved five tracks arranged in a plus-shape, with spacing 500 μm. The center track was located 3.5 mm anterior and 2.5 mm lateral to bregma, and all tracks extended from 4.3 to 5.7 mm ventral to bregma. In each track, 15 injections of 23 nL were made at 100 μm intervals, pausing for ten seconds between injections, and for 1 min at the bottom of each track. In total 1.7 μl of virus were delivered to each hemisphere over a period of about 20 min.

Rats recovered for a minimum of one week, with ad lib access to food and water, before returning to training. Rats with virus injections returned to training, but did not begin inactivation experiments until a minimum of 6 weeks had passed, to allow for virus expression.

## Optogenetic perturbation experiments

During inactivation experiments, rats performed the task in a behavioral chamber outfitted with a dual fiber optic patch cable connected to a splitter and a single-fiber commutator (Princetel) mounted in the ceiling. This fiber was coupled to a 200 mW 532 nm laser (OEM Laser Systems) under the control of a mechanical shutter (ThorLabs) by way of a fiber port (ThorLabs). The laser power was tuned such that each of the two fibers entering the implant received between 25 and 30 mW of light when the shutter was open.

Each rat received several sessions in which the shutter remained closed, in order to acclimate to performing the task while tethered. Once the rat showed behavioral performance while tethered that was similar to his performance before the implant surgery, inactivation sessions began. During these sessions, the laser shutter was opened (causing light to flow into the implant, activating the eNpHR3.0 and silence neural activity) on 7% of trials each in one of three time periods. 'Outcome period' inactivation began when the rat entered the bottom center port at the end of the trial, and ended either when the rat had left the port and remained out for a minimum of 500ms, or after 2.5 s. 'Choice period' inactivation began at the end of the outcome period and lasted until the rat entered the choice port on the following trial. 'Both period' inactivation encompassed both the outcome period and the choice period. The total duration of the inactivation therefore depended in part on the movement times of the rat, and was somewhat variable from trials to trial (*Figure 5—figure supplement 2*). If a scheduled inactivation would last more than 15 s, inactivation was terminated, and that

trial was excluded from analysis. Due to constraints of the bControl software, inactivation was only performed on even-numbered trials.

## Analysis of optogenetic effects on behavior

We quantify the effects of optogenetic inhibition on behavior by computing separately the planning index for trials following inactivation of each type (outcome period, choice period, both periods) and for control trials. Specifically, we fit the trial history regression model of *Equation 1* with a separate set of weights for trials following inactivation of each type:

$$log\left(\frac{P_{left}(t)}{P_{right}(t)}\right) = \sum_{\tau=1}^{T} \beta_{CR,i}(\tau) \cdot CR(t-\tau) + \sum_{\tau=1}^{T} \beta_{CO,i}(\tau) \cdot CO(t-\tau) +$$
$$\sum_{\tau=1}^{T} \beta_{UR,i}(\tau) \cdot UR(t-\tau) + \sum_{\tau=1}^{T} \beta_{UO,i}(\tau) \cdot UO(t-\tau)$$

(15)

$$i = \begin{cases} cntrl, \text{trail } t-1 \text{ was control trail} \\ out, \text{trail } t-1 \text{ had outcome period inactivation} \\ ch, \text{trail } t-1 \text{ had choice period inactivation} \\ both, \text{trail } t-1 \text{ had both} \end{cases}$$

(16)

We used maximum a posteriori fitting in which the priors were Normal(0,1) for weights corresponding to control trials, and Normal($\beta_{X, cntrl}$, 1) for weights corresponding to inactivation trials, where $\beta_{X, cntrl}$ is the corresponding control trial weight – e.g. the prior for $\beta_{CR, out}(1)$ is Normal($\beta_{CR, cntrl}(1)$, 1). This prior embodies the belief that the effect of inactivation on behavior is likely to be small, and that the direction of any effect is equally likely to be positive or negative. This ensures that our priors cannot induce any spurious differences between control and inactivation conditions into the parameter estimates. We then compute a planning index separately for the weights of each type, modifying *Equation 3*:

$$PlanningIndex_i(\tau) = [\beta_{CR,i}(\tau) - \beta_{UR,i}(\tau)] + [\beta_{UO,i}(\tau) - \beta_{CO,i}(\tau)]$$

(17)

We compute the relative change in planning index for each inactivation condition: (*PlanningIndex$_i$* - *PlanningIndex$_{cntrl}$*) / *PlanningIndex$_{cntrl}$*, and report three types of significance tests on this quantity. First, we test for each inactivation condition the hypothesis that there was a significant change in the planning index, reporting the results of a one-sample t-test over rats. Next, we test the hypothesis that different inactivation conditions had effects of different sizes on the planning index, reporting a paired t-test over rats. Finally, we test the hypothesis for each condition that inactivation had a different effect than sham inactivation (conducted in rats which had not received virus injections to deliver eNpHR3.0), reporting a two-sample t-test.

To test the hypothesis that inactivation specifically impairs the effect of distant past outcomes on upcoming choice, we break down the planning index for each condition by the index of the weights the contribute to it:

$$PlanningIndex_i(\tau) = [\beta_{CR,i}(\tau) - \beta_{UR,i}(\tau)] + [\beta_{UO,i}(\tau) - \beta_{CO,i}(\tau)]$$

(18)

We report these trial-lagged planning indices for each inactivation condition, and assess the significance of the difference between inactivation and control conditions at each lag using a paired t-test.

## Synthetic datasets

To generate synthetic datasets for comparison to optogenetic inactivation data, we generalized the behavioral model to separate the contributions of representations of expected value and of immediate reward. In particular, we replaced the learning equation within the model-based RL agent (*Equation 7*) with the following:

$$V(o) \leftarrow \begin{cases} \alpha_{value} \cdot V(o) + \alpha_{reward} \cdot R_t + (1 - \alpha_{value} - \alpha_{reward}) \cdot E[V], \text{for } o = o_t \\ \alpha_{value} \cdot V(o) - \alpha_{reward} \cdot R_t + (1 - \alpha_{value} - \alpha_{reward}) \cdot E[V], \text{for } o \neq o_t \end{cases}$$

(19)

where $\alpha_{value}$ and $\alpha_{reward}$ are separate learning rate parameters, constrained to be nonnegative and to have a sum no larger than one, and E[$V$] represents the expected reward of a random-choice policy on the task, which in the case of our task is equal to 0.5.

To generate synthetic datasets in which silencing the OFC impairs choice value representations, outcome value representations, or reward representations, we decrease the parameter $\beta_{plan}$, $\alpha_{value}$, or $\alpha_{reward}$, respectively. Specifically, we first fit the model to the dataset for each rat in the optogenetics experiment (n=9) as above (i.e. using *equation 7* as the learning rule) to obtain maximum a posteriori parameters. We translated these parameters to the optogenetics (*equation 18*) version of the model by setting $\alpha_{value}$ equal to the fit parameter $\alpha$ and $\alpha_{reward}$ equal to 1 - $\alpha$. We then generated four synthetic datasets for each rat. For the control dataset, the fit parameters were used on trials of all types, regardless of whether inhibition of OFC was scheduled on that trial. For the 'impaired outcome values' dataset, $\alpha_{value}$ was decreased specifically for trials with inhibition scheduled during the outcome period or both periods, but not on trials with inhibition during the choice period or on control trials. For the 'impaired reward processing' dataset, $\alpha_{reward}$ was decreased on these trials instead. For the 'impaired decision-making' dataset, $\beta_{plan}$ was decreased specifically on trials following inhibition. In all cases, the parameter to be decreased was multiplied by 0.3, and synthetic datasets consisted of 100,000 total trials per rat.

## Histological verification of targeting

We verified that surgical implants were successfully placed in the OFC using standard histological techniques. At experimental endpoint, rats with electrode arrays were anesthetized, and microlesions were made at the site of each electrode tip by passing current through the electrodes. Rats were then perfused transcardially with saline followed by formalin. Brains were sliced using a vibratome and imaged using an epifluorescent microscope. Recording sites were identified using these microlesions and the scars created by the electrodes in passing, as well dimples in the surface of the brain. Locations of optical fibers were identified using the scars created by their passage. Location of virus expression was identified by imaging the YFP conjugated to the eNpHR3.0 molecule *Figure 5—figure supplement 1*.

Data collected for the purpose of this paper will be posted on Figshare upon acceptance. Software used to analyze the data will be made available as a Github release. Software used for training rats and design files for constructing behavioral rigs are available on the Brody lab website.

## Acknowledgements

We thank Athena Akrami for assistance with array implant surgeries; Jovanna Teran, Klaus Osorio, Adrian Sirko, Samantha Stein, and Lillianne Teachen for assistance with animal training; and Peter Bibawi for help with histology. We thank Luca Mazzucato for identifying an error in an early version of our data processing pipeline. We thank Nathaniel Daw, Yael Niv, Geoffrey Schoenbaum, and Thomas Akam for helpful discussions, and Athena Akrami, Christine Constantinople, Nathaniel Daw, Cristina Domnisoru, Jeff Gauthier, Chuck Kopec, Olga Lositsky, Marcelo Mattar, Bas van Opheusden, Angela Radulescu, Ben Scott, Kim Stachenfeld, and Bob Wilson for helpful comments on the manuscript. KJM was supported by training grant NIH T-32 MH065214 to the Princeton Neuroscience Institute, and by a Harold W Dodds fellowship from Princeton University.

## Additional information

### Funding

| Funder | Grant reference number | Author |
|---|---|---|
| National Institutes of Health | T-32 MH065214 | Kevin J Miller |
| Princeton University | Harold W Dodds Fellowship | Kevin J Miller |

| Funder | Grant reference number | Author |
|--------|------------------------|--------|

The funders had no role in study design, data collection and interpretation, or the decision to submit the work for publication.

## Author contributions

Kevin J Miller, Conceptualization, Data curation, Formal analysis, Investigation, Visualization, Methodology, Writing – original draft, Writing – review and editing; Matthew M Botvinick, Conceptualization, Supervision, Funding acquisition, Writing – review and editing; Carlos D Brody, Conceptualization, Resources, Supervision, Funding acquisition, Writing – original draft, Writing – review and editing

## Author ORCIDs

Kevin J Miller ⓘ http://orcid.org/0000-0002-3465-2512
Carlos D Brody ⓘ http://orcid.org/0000-0002-4201-561X

## Ethics

All experimental procedures were performed in strict accordance with the recommendations in the Guide for the Care and Use of Laboratory Animals of the National Institutes of Health, and were approved by the Princeton University Institutional Animal Care and Use Committee (protocol #1853).

## Decision letter and Author response

Decision letter https://doi.org/10.7554/eLife.64575.sa1
Author response https://doi.org/10.7554/eLife.64575.sa2

# Additional files

## Supplementary files

• Transparent reporting form

## Data availability

Data collected for the purpose of this paper are available on FigShare at https://doi.org/10.6084/m9.figshare.20449140. Software used to analyze the data is available on Github at https://github.com/kevin-j-miller/MBB2022-orbitofrontal-learning-choosing/ copy archived at swh:1:rev:c65b4ca63e73927acbac07168f220741a02a4301. Software used for training rats and design files for constructing behavioral rigs are available at https://github.com/kevin-j-miller/MBB2022-orbitofrontal-learning-choosing/tree/main/bControl_protocols.

The following datasets were generated:

| Author(s) | Year | Dataset title | Dataset URL | Database and Identifier |
|-----------|------|---------------|-------------|-------------------------|
| Miller KJ | 2022 | Data from Value Representations in the Rodent Orbitofrontal Cortex Drive Learning, not Choice | https://doi.org/10.6084/m9.figshare.20449140 | figshare, 10.6084/m9.figshare.20449140 |
| Miller KJ | 2022 | Code from Value Representations in the Rodent Orbitofrontal Cortex Drive Learning, not Choice | https://doi.org/10.5281/zenodo.6974148 | Zenodo, 10.5281/zenodo.6974148 |

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
