## [Editor Report]

In this manuscript, Miller et al., use the two-step task, a task initially designed to discern model-free from model-based behavior, to probe OFC at specific times throughout the two stages of the task to understand OFC's role in choice and learning. The authors exploited an interesting feature of the two-step task that allows choice and learning to be separated into separate stages. They then used this feature to clearly show that OFC is not necessary for choice behavior in a well-learned choice task, although it is required for updating the model based on trial-by-trial reward information. The ability to separate learning from choice in a decision-making task is a unique and novel approach for probing OFC function, making this manuscript an important contribution to our understanding of OFC function.

---

## [Decision Letter]

**Decision letter after peer review:**

Thank you for submitting your article "Value Representations in the Rodent Orbitofrontal Cortex Drive Learning, not Choice" for consideration by *eLife*. Your article has been reviewed by 3 peer reviewers, and the evaluation has been overseen by Geoffrey Schoenbaum as the Reviewing Editor and Kate Wassum as the Senior Editor. The following individual involved in review of your submission has agreed to reveal their identity: Peter Rudebeck (Reviewer #2).

The reviewers have discussed the reviews with one another and the Reviewing Editor has drafted this decision to help you prepare a revised submission.

Summary:

In this manuscript, Miller et al., use the two-step task, a task initially designed to discern model-free from model-based behavior, to probe OFC at specific times throughout the two stages of the task to understand OFC's role in choice and learning. The authors exploited an interesting feature of the two-step task that allows choice and learning to be separated into separate stages. They then used this feature to clearly show that OFC is not necessary for choice behavior in a well-learned choice task, although it is required for updating the model based on trial-by-trial reward information. The ability to separate learning from choice in a decision-making task is a unique and novel approach for probing OFC function, making this manuscript an important contribution to our understanding of OFC function.

Revisions:

In the discussion and the reviews below, each reviewer was enthusiastic about the results and felt they should be published without undue delay. While there were a number of questions raised, in the discussion it was felt that three key areas in particular were particularly important or essential to address in the revision. As the authors will see, these general areas encompass many of the specific requests in each of the reviews.

(1) It was felt that it was important for the authors to provide more information and specification of the modeling, so that the details would be more accessible to readers.

(2) Additional details regarding the single unit results, and possibly changes to bring it more into alignment with the epochs employed in the inactivation experiment, are necessary.

(3) Some softening of the hard claim that OFC is not necessary for choice; while the results of this specific study are clear, the possibility exists that in other settings this may not be true. This is discussed a bit in the current paper, but this issue was raised in each review and also in the discussion in various ways. These thoughts should be addressed in the revision.

*Reviewer #1 (Recommendations for the authors):*

In this manuscript, Miller et al., further dissect OFC's role in the two-step task, a task initially designed to discern model-free from model-based behavior. While their previous study showed that session-wide pharmacological inactivation of the OFC shifts rats to learn from rewards in a model-free way, they found that such full-session inactivation played no role in impairing choice behavior on individual trials. Here they used optogenetics to probe OFC at specific times throughout the two stages of the task to further understand OFC's role in choice and learning. Inactivation of OFC during the initial choice phase had no effect on behavior, although inactivation during the second phase, as well as across both phases, impaired learning of probabilistic outcomes. Comparing their behavioral results to a well-fit hybrid RL model of behavior, they showed that this change in the rats' behavior best corresponds with a loss of expected outcome value rather than an impairment of choice value. Electrophysiology recordings in OFC revealed that expected outcome value was substantially better coded for during the task than chosen value, consistent with the inactivation results.

The authors exploited an interesting feature of the two-step task that allows choice and learning to be separated into separate stages. They then used this feature to clearly show that OFC is not necessary for choice behavior in a well-learned choice task, although it is required for updating the model based on trial-by-trial reward information. The ability to separate learning from choice in a decision-making task is a unique and novel approach for probing OFC function, making this manuscript an important contribution to our understanding of OFC function.

Overall I think this is a terrific study. The approach is novel and creative, taking advantage of a unique task design, combined with unit recording and temporally specific manipulations, to put at risk an important set of hypotheses about an important brain region – the OFC. The paper is also exceptionally well-written and, to the authors credit, I think they fairly consider the array of results both in setting up the importance of their study and in understanding the significance of the results. I definitely have questions, but I do not mean them to detract from my enthusiastic support.

So one question I had was whether the task favors the result that they obtained. Specifically, it seems to me that prior work, starting with the original Daw report and including rodent work, shows that choice in this setting is driven by a combination of MB and MF information. Or at least that both types of value can support many of the choices, whereas the learning in the critical trials requires MB information. I think this leads to an asymmetry where choice is less dependent on MB information than the type of learning that is isolated; while perhaps this can be mitigated by which trials are analyzed, I wonder if this asymmetry might lead to some of the results. That is, the finding that OFC activity is only weakly related to and unnecessary for choice in the task could reflect the weaker dependence of choice on MB information. And vice versa for the involvement in learning. If the task were more symmetrical – as one might speculate probe test behavior in devaluation is for example – could a stronger role for OFC in driving choices appear? Note this would not affect the significance of the current results for settings like economic choice, at least as experimentally tested; it would simply be a weaker distinction I think in the function. In any event, the authors could comment on this; if they disagree with the idea of an asymmetry at least in the analysis, then a careful explanation of that to me and perhaps in the paper would be useful. Or perhaps this is what the authors are thinking? I was confused a bit where they note several times that some other brain area is functioning as a controller for behavior – do they mean a MF area like striatum? Or another MB controller?

Another question I have is how the authors think the update might occur. Specifically, do the authors suggest that this is in agreement with the credit assignment hypothesis, indicating that these are incremental changes in values of the second stage states? Or could this be modeled as updates in state transitions if the final outcome is modeled as states with different probabilities of reward. These possibilities might not be distinguishable with the task, but perhaps they should be mentioned as they pertain to different interpretations of OFC function.*Reviewer #2 (Recommendations for the authors):*

This manuscript has the feel of one that has been well reviewed before and so my suggestions are really just to make the paper more accessible:

The computational modeling is really under specified/described in the paper. I realize that the authors have published the model elsewhere but without more information in this manuscript a reader is left a little unsure how to interpret the effects reported. The authors need to fix this lack of specificity in two places: (1) in the behavioral analyses of the data and (2) the modeling of the effects of optogenetic inhibition. In both places the model is introduced rather suddenly. Figures 1 and 4 have schematics for how the model generally works but more of the text should be dedicated to explaining how the model is implemented. Similarly, in the section analyzing the effects of optogenetic inhibition on behavior the section/paragraph using modeling is quite opaque. It is simply introduced as an "artificial planning agent" but without specifying how this works a reader is unable to evaluate this approach. The authors should take the time to explain in the text how they simulated the choices of this agent and how that in turn was used to generate the different effects on behavior. This is essential to ensuring that the sophisticated approach to modeling behavior that was taken here can be fully appreciated.

The analyses of the neurophysiology data are solid but lack detail. The authors should provide: (1) breakdowns of the numbers of neurons/MUAs recorded per rat. (2) Given the variability in regression coefficients between animals, the authors should also ensure that their neurophysiology effects are not simply being driven by a single or subset of animals. This could be done by adding subject as a random effect into their models, subject dropping, data splitting or similar. Either way, the authors should provide evidence that the effects are not solely driven by a single/few animals. Related to the above, the authors should also prove that this effect of animal is consistent for both the CPD analyses as well as the number of neurons signaling the different factors as shown in figure 3 – supplement 3.

While the analyses of behavior using the computational model is really elegant, only presenting the change in planning index as the primary measure of change in behavior after OFC inactivation feels very far removed from the data. The authors should provide more information in both text and figures on how inactivation of OFC alters subsequent choices or other metrics of behaviors such as reaction time. The authors could consider showing change in regression coefficients similar to supplemental Figure 1, win-stay/loose-shift behaviors etc. Further, analyses of reaction times/time to initiate choices after optogenetic inhibition could be another way to get insight into the effects of inactivating OFC. On trials after laser delivery, do rats slow down on the first or second stage of the two-step task? If they are slower to choose between left and right on the first step this might indicate a change in policy/decision-making whereas if they are slower to initiate/move based on the second stage it would indicate altered value expectations.*Reviewer #3 (Recommendations for the authors):*

The central question in this study is whether value representations in the (rodent) OFC drive choice or whether they drive learning. The core question is an important one in neuroscience because the OFC is the focus of a great deal of long-standing debate. The authors of the study separate these possibilities using a well-known multi-step task. The task is powerful because it provides a computationally strong and elegant way to separate these things. Note that one innovative feature of the authors' research program is training rodents to perform this task – they have done so in previous work, and, here leverage that advance to begin to answer neuroscientific questions. Formally speaking, the authors use two complementary methods, unit recordings and optogenetic silencing. Results from both methods wind up pointing in the same direction, towards the learning account and away from the choice account.

I have a few thoughts that relate more to the presentation than to the data. I wonder how the frame structure of learning vs. choice relates to the existing larger literature on OFC. From my understanding, this frame skips cognitive mapping functions and the related expectancy signaling accounts, and also skips accounts in which OFC helps navigate complex of model-based choices. Im not saying that these accounts are all in conflict or must be addressed, but most readers will be familiar with them, and the authors should work through how they all relate. Are they saying these others are all part of a more general learning-based account? Additional explanation would help.

Relatedly, it would seem that the null hypothesis here is that OFC does both learning and choice – why not?? Is there any a priori reason it shouldn't do them? It seems reductive to insist a brain region can only have one function.

So for example, the authors find that OFC is more active for outcomes than for offers. But it is still active for offers. One might therefore conclude most parsimoniously that OFC is involved more strongly in learning, but still involved in choice as well. Indeed, I would look at the significant CPDs for choice variables and say the authors have presented clear evidence for both. It's not even clear that bigger CPDs for outcome than offer implies that OFC is more involved in learning than choice – there are several reasons you could imagine why you would see this effect (e.g. learning is more difficult, requires more attention, is more narrowly focused on OFC). These other possibilities are all consistent with the "both" theory, which the authors do not seem to consider. Like, these sentences, for example, seem to contain within them the assumption that weak but significant correlations are otiose… which seems unlikely…. "We find that neurons in the OFC correlate with the expected values of the items being learned about, but only weakly with the expected values of the items being chosen between. This indicates that, within a repeated-trials task, neural representations in the OFC carry value information that is selective for a role in learning, but do not seem to carry value information selective for a role in choice." Moreover, the analysis is conservative – if I understand correctly it attributes all variance to other confounding variables and counts only remaining variance. I see why the authors would do it, but this would seem to artificially deflate responses. That itself might not be bad, if we keep it in mind when interpreting the data. But the result here is that the effects are smaller than expected.

Even so, the effect size numerically is not as big as the authors' rhetoric implies – 1.1% is the good number and 0.52% is the bad number. Is it so obvious that we should interpret these as the authors want us to – that 1.1% means OFC is all about learning and 0.52% means it has nothing to do with choice? Stronger results are reported a few sentences later (the ratio is 3.8 or 4.0). However, again, I think it is a step too far to consider this to be evidence that OFC does one thing and not another. I am reminded, for example, of the fact that nodulation by attention in visual area V4 and strong driving by visual stimuli – in those areas, the modulation ratio is about 5:1. That doesn't mean, however, that V4 doesn't participate in attention.

For the causal manipulation, I think a clearer case can be made about its novelty. The issue is a series of papers by Gardner and Schoenbaum using causal/optogenetic approaches to demonstrate the relevance of OFC to learning and learning-like processes. The authors here say that "Our results build on this by pinpointing, out of all those possibilities, a specific computational role for OFC: it provides outcome value estimates to a learning system, elsewhere in the brain, that updates the estimates of these values." However, again, this claim is not as strong as it seems – there is no reason given to think that OFC only has one role or that its role in this task is the only role that it plays in cognition more generally. Reading this paper along with the Gardner papers, I would say that the authors' conclusion is wrong in that they do not reject alternative roles for OFC other than choice.

So what are we left with here? I believe the authors that learning-related activity is stronger than choice-related activity. And I think it's worth pointing out that this suggests OFC plays a more important role in learning than choice, although it's not necessarily further than a suggestion. The optogenetic silencing results are stronger, but the novelty of those results – especially relative to Schoenbaum's series of papers – is not clear.

---

## [Author Response]

Revisions:In the discussion and the reviews below, each reviewer was enthusiastic about the results and felt they should be published without undue delay. While there were a number of questions raised, in the discussion it was felt that three key areas in particular were particularly important or essential to address in the revision. As the authors will see, these general areas encompass many of the specific requests in each of the reviews.(1) It was felt that it was important for the authors to provide more information and specification of the modeling, so that the details would be more accessible to readers.

Thank you for this comment, we have added substantially more detail on the modeling and behavior analysis, to clarify them, in three places. The first is a new figure (Figure 2) with panels on the cognitive model and one the trial-history regression analysis. The second is a substantially expanded description in the section of Results that is devoted to behavior (“Planning Strategy in the Two-Step Task Separates Choosing and Learning”). The third is an expanded description of the synthetic inactivation experiments and analysis of synthetic data in the section of Results describing optogenetics experiments (“Inactivations of OFC impair update process, not choice process”). We hope this will successfully make the modeling more accessible to readers.

(2) Additional details regarding the single unit results, and possibly changes to bring it more into alignment with the epochs employed in the inactivation experiment, are necessary.

Following the individual reviewers’ requests, we have added additional detail regarding the electrophysiological recordings in several new supplemental figures. Two of these focus on robustness of results across rats (Figure 4-S4: ephys CPD data analysis separated by rat; 4-S5: ephys fraction of significant units separated by rat), as requested by R2. A third (Figure 4-S6) addresses the fact that CPD is a conservative measure when regressors are correlated, a concern raised by R3, by removing a regressor (interaction between outcome port and reward) that is highly correlated with choice-related value regressors. The results remain substantially the same, thus buttressing the analysis and conclusions in the main text. The last addition we made follows a request from R1 and adds a new analysis that addresses the fact that choices in our task are not driven exclusively by expected value, by using choice-related regressors that consider not just value but the total decision variable (Figure 4-S7).

We have also added additional detail clarifying how the epochs in the inactivation experiment relate to the timebins of analysis in the electrophysiology experiment. In Figure 5-S2, we show a timecourse of the probability distributions of being in each optogenetics epoch, aligned in the same way as our analysis of the electrophysiology data.

(3) Some softening of the hard claim that OFC is not necessary for choice; while the results of this specific study are clear, the possibility exists that in other settings this may not be true. This is discussed a bit in the current paper, but this issue was raised in each review and also in the discussion in various ways. These thoughts should be addressed in the revision.

We have softened these claims throughout the paper, and clarified the description in our discussion.

Reviewer #1 (Recommendations for the authors):Overall I think this is a terrific study. The approach is novel and creative, taking advantage of a unique task design, combined with unit recording and temporally specific manipulations, to put at risk an important set of hypotheses about an important brain region – the OFC. The paper is also exceptionally well-written and, to the authors credit, I think they fairly consider the array of results both in setting up the importance of their study and in understanding the significance of the results. I definitely have questions, but I do not mean them to detract from my enthusiastic support.

We are very grateful for the reviewer’s enthusiastic support!

So one question I had was whether the task favors the result that they obtained. Specifically, it seems to me that prior work, starting with the original Daw report and including rodent work, shows that choice in this setting is driven by a combination of MB and MF information. Or at least that both types of value can support many of the choices, whereas the learning in the critical trials requires MB information. I think this leads to an asymmetry where choice is less dependent on MB information than the type of learning that is isolated; while perhaps this can be mitigated by which trials are analyzed, I wonder if this asymmetry might lead to some of the results. That is, the finding that OFC activity is only weakly related to and unnecessary for choice in the task could reflect the weaker dependence of choice on MB information. And vice versa for the involvement in learning. If the task were more symmetrical – as one might speculate probe test behavior in devaluation is for example – could a stronger role for OFC in driving choices appear? Note this would not affect the significance of the current results for settings like economic choice, at least as experimentally tested; it would simply be a weaker distinction I think in the function. In any event, the authors could comment on this; if they disagree with the idea of an asymmetry at least in the analysis, then a careful explanation of that to me and perhaps in the paper would be useful. Or perhaps this is what the authors are thinking? I was confused a bit where they note several times that some other brain area is functioning as a controller for behavior – do they mean a MF area like striatum? Or another MB controller?

We thank the reviewer for bringing up this important question. The concern is that there may be an asymmetry between learning and choosing in terms of how “model-based” the rats are. In this situation, learning about outcome port values may be uniquely the purview of a model-based system, while deciding between choice ports might be influenced by other systems, including a model-free one that our analyses do not focus on.

We first note that while the reviewer is correct that most versions of the two-step task show an apparent influence of model-free RL on behavior, our rat version is unusual in that it apparently does not. Specifically, if we add a model-free agent to our mixture-of-agents behavioral model, we find that this does not meaningfully improve quality of fit. In our revision, we now make this point more robustly, comparing several different types of model-free agent (TD(1), TD(0), and TD(λ)), moving this analysis into a main-text figure (Figure 2c), and adding the following text to the Results section:

“Among these additional components we tested were several model-free reinforcement learning strategies, which have been reported to contribute to behavior on similar multi-step tasks (Daw et al., 2011; Hasz and David Redish, 2018; Dezfouli and Balleine, 2019; Groman et al., 2019; Akam et al., 2020; Miranda et al., 2020). That adding them to our model does not improve quality of fit suggests that model-free reinforcement learning does not contribute meaningfully to rat behavior in our task. This is important for our analysis because these strategies involve value representations of their own – that they are not part of our model means that model-based value representations are the only representations of expected reward that are present.”

This does not eliminate the possibility of an asymmetry though, because our rat behavior does show an apparent influence of other processes – perseveration, bias, and one we call “novelty preference”. These processes do not involve representations of value in the sense of expected future reward (they are entirely insensitive to reward). However they do influence choice, and their representations might therefore be considered “value” in the sense of revealed preference. In our revision, we have added a new analysis of neural recordings that uses analogs of “choice value difference” and “chosen value” that consider the combined influence of all of these components, instead of isolating the contribution of the model-based component, and added the following text to our Results section:

“These results were robust to replacing the choice-related value regressors with analogs that consider the full decision variable, including contributions from perseveration and novelty preference as well as expected value (Figure 4-S5).”

We have also added a new analysis of our optogenetics quantifying any effect of inactivation on perseverative, model-free, or novelty preference patterns of behavior, and added the following text to our Results section:

As in our previous study, inactivation of OFC did not significantly affect other regression-based behavioral indices (Figure 5-S2).

Together, we believe that these findings indicate that it is unlikely for our results to be driven by an asymmetry between learning vs. choosing in terms of how “model-based” vs “model-free” each is, because in this task, rat behavior appears to have no significant model-free component.

Another question I have is how the authors think the update might occur. Specifically, do the authors suggest that this is in agreement with the credit assignment hypothesis, indicating that these are incremental changes in values of the second stage states? Or could this be modeled as updates in state transitions if the final outcome is modeled as states with different probabilities of reward. These possibilities might not be distinguishable with the task, but perhaps they should be mentioned as they pertain to different interpretations of OFC function.

Our data are consistent with the idea that the OFC supports incremental learning, either of values of second stage states or of state transition probabilities. This distinction is closely related to the distinction between “common currency” vs “specific outcome” expectancy signals brought up by R3. We have added a paragraph to the discussion clarifying this:

“Our results are consistent with the view that the OFC carries expectancy signals (Schoenbaum and Roesch, 2005; Rudebeck and Murray, 2014) indicating which outcomes are expected to follow from the current state. This view is distinct from the “chosen value” view described earlier in that it claims that these signals indicate the particular identities of the expected outcomes (e.g. different types of foods) rather than the abstract “reward” or “common currency” value found in reinforcement learning or neuroeconomic theories. Our experiment does not speak to the difference between these views, because the only reward available in our task is a water droplet of a fixed size. Our “outcome value” correlates might therefore reflect expectations of this reward (the water droplet) in particular, and play a role in updates based on this expectation. In formal models, this might be described as updating estimates of a state transition probability (e.g. from an “at the outcome port” state to a “receive water droplet” state). Alternatively, “outcome value” might abstract over many different possible rewarding outcomes. In formal models, this might be described as updating estimates of the reward function (e.g. in an “at the outcome port” state).”

Reviewer #2 (Recommendations for the authors):This manuscript has the feel of one that has been well reviewed before and so my suggestions are really just to make the paper more accessible:The computational modeling is really under specified/described in the paper. I realize that the authors have published the model elsewhere but without more information in this manuscript a reader is left a little unsure how to interpret the effects reported. The authors need to fix this lack of specificity in two places: (1) in the behavioral analyses of the data and (2) the modeling of the effects of optogenetic inhibition. In both places the model is introduced rather suddenly. Figures 1 and 4 have schematics for how the model generally works but more of the text should be dedicated to explaining how the model is implemented. Similarly, in the section analyzing the effects of optogenetic inhibition on behavior the section/paragraph using modeling is quite opaque. It is simply introduced as an "artificial planning agent" but without specifying how this works a reader is unable to evaluate this approach. The authors should take the time to explain in the text how they simulated the choices of this agent and how that in turn was used to generate the different effects on behavior. This is essential to ensuring that the sophisticated approach to modeling behavior that was taken here can be fully appreciated.

We thank the reviewer for pointing this out. We agree that it is important for the manuscript to be clear in the Results section about the behavioral model and how it contributes to these analyses. We have addressed this by changing it in two places. The first is in the behavioral Results section, which we have expanded to include descriptions of the cognitive model and the trial-history regression analysis. We have combined the figure panels that go with this section into a new main text figure (Figure 2). The relevant paragraphs now read:

“As in our previous study (Miller, Botvinick and Brody, 2017), rat behavior was well-described by a cognitive model combining this model-based planning strategy with a mixture of three additional components. The first of these is “perseveration”, which reflects a tendency to repeat past choices, regardless of their outcomes (Akaishi et al., 2014; Miller, Shenhav and Ludvig, 2019). The second, which we term “novelty preference”, reflects a tendency to repeat (or to switch away from) choices that lead to an uncommon transition, regardless of whether or not they are rewarded. The third is a constant side bias, reflecting an overall tendency to prefer either the right or the left choice port. Each of these components is associated with a weighting parameter (*β*), reflecting the strength of its influence on the decision between the left and right choice port on each trial (Figure 2a, right: “Choose a choice port”). Fitting these parameters to the dataset for each rat, we find that the planning and perseverative components earn weights that are large and positive (Figure 2b), while the novelty preference and bias components earn weights that are generally smaller and differ in sign among rats (Figure 2b). The planning and perseveration components are each associated with a learning rate parameter (*α*), which reflects the relative influence of trials in the recent past vs the more distant past. Learning rates for the planning component were consistently larger than those for the perseverative component (Figure 2-S1).

We validate that our cognitive model provides a good description of rat behavior in two different ways. The first way is quantitative model comparison: we compute a quality-of-fit score for the model using cross-validated likelihood, and compare this score between our model and various alternatives (Figure 2c). Alternative models which are missing any of the four components perform substantially worse (Figure 2c, red points), while alternative models adding various additional components do not perform substantially better (Figure 2c, green points). Among these additional components we tested were several model-free reinforcement learning strategies, which have been reported to contribute to behavior on similar multi-step tasks (Daw et al., 2011; Hasz and David Redish, 2018; Dezfouli and Balleine, 2019; Groman et al., 2019; Akam et al., 2020; Miranda et al., 2020). That adding them to our model does not improve quality of fit suggests that model-free reinforcement learning does not contribute meaningfully to rat behavior in our task. This is important for our analysis because these strategies involve value representations of their own – that they are not part of our model means that model-based value representations are the only representations of expected reward that are present.

Our second way of validating the cognitive model makes use of a trial-history regression analysis (Lau and Glimcher, 2005), which provides a theory-neutral way of characterizing the patterns present in a behavioral dataset. This analysis fits separate weights for each of the four possible outcome types (common-rewarded, common-omission, uncommon-rewarded, uncommon-omission). These weights will be positive if the rat tends to repeat choices that lead to that outcome, and negative if the rat tends to switch away from such choices. Behavioral datasets produced by different strategies will have different patterns of weights (Miller, Brody and Botvinick, 2016). For example, a model-based planning strategy will show more positive weights for common-reward than for uncommon-reward (since a rewarding outcome port visited after a common transition is likely to be reached again by repeating the choice, while one reached after an uncommon transition is more likely to be reached by switching to the other choice port instead), as well as more negative weights for common-omission than for uncommon-omission (since a unrewarding outcome port visited after a common transition is likely to be avoided by switching the choice, while one reached after an uncommon transition is more likely to be avoided by repeating the choice). As in our previous study, rats trained for the present study universally show this qualitative pattern, and the quantitative patterns present in their weights are well-matched by fits of the cognitive model (example rat: Figure 2d; all rats: Figure 2-S2). To summarize the pattern and compare it to others in behavior, we define a “planning index” as the linear combination of weights consistent with the planning strategy, as well as a “stay” index, a “common-stay/uncommon-switch” index, and a “win-stay/lose-switch” index quantifying other patterns (Figure 2e).[…] All rats showed large values of the planning index and the stay index, and much smaller values of the common-stay/uncommon-switch and win-stay/lose-switch indices (Figure 2f).”

The second is in the optogenetics Results section. We have expanded our description of the modeling here, both in terms of how we generate the synthetic datasets and how we analyze them. We have also added a new equation to the main text. The relevant passage now reads:

“To help understand which aspect of the behavior was affected by silencing the OFC, we used our cognitive model (Figure 2a) to perform three different types of synthetic inactivation experiments. […] This is because value acts as a summarized memory of previous trials’ outcomes, and attenuating it affects the influence of all of these.”

The analyses of the neurophysiology data are solid but lack detail. The authors should provide: (1) breakdowns of the numbers of neurons/MUAs recorded per rat. (2) Given the variability in regression coefficients between animals, the authors should also ensure that their neurophysiology effects are not simply being driven by a single or subset of animals. This could be done by adding subject as a random effect into their models, subject dropping, data splitting or similar. Either way, the authors should provide evidence that the effects are not solely driven by a single/few animals. Related to the above, the authors should also prove that this effect of animal is consistent for both the CPD analyses as well as the number of neurons signaling the different factors as shown in figure 3 – supplement 3.

We have added two new supplemental figures (Figure 4-S4 and 4-S5). The first of these contains both a breakdown of the number of singles and multis recorded in each rat, as well as an analysis which uses a data splitting approach to address the concern that effects may be driven by only a subset of animals. Specifically, we compute the timecourse of CPDs (as in main text Figure 4b and c) separately for each rat. The key findings that outcome value earns a higher population-level CPD than either type of choice-related value is visible in each of the six animals. The second new figure computes the fraction of units which significantly encode each regressor in each time bin, separately for each rat (as in Figure 4-S1). The finding that outcome value is encoded by a larger fraction of units than either type of choice value information is visible in five of the six rats (the sixth has only a small number of recorded units, few of which encode any of the three expected value regressors).

While the analyses of behavior using the computational model is really elegant, only presenting the change in planning index as the primary measure of change in behavior after OFC inactivation feels very far removed from the data. The authors should provide more information in both text and figures on how inactivation of OFC alters subsequent choices or other metrics of behaviors such as reaction time. The authors could consider showing change in regression coefficients similar to supplemental Figure 1, win-stay/loose-shift behaviors etc. Further, analyses of reaction times/time to initiate choices after optogenetic inhibition could be another way to get insight into the effects of inactivating OFC. On trials after laser delivery, do rats slow down on the first or second stage of the two-step task? If they are slower to choose between left and right on the first step this might indicate a change in policy/decision-making whereas if they are slower to initiate/move based on the second stage it would indicate altered value expectations.

We have added new supplemental figures showing in more detail how inactivation affects behavioral strategy. The first of these (Figure 5-S2) shows the effect of inactivation on all four behavioral indices (which are now described in Figure 2). Only the planning index is significantly affected. Figure 5-S3 breaks this down further by lag, and figure 5-S4 by individual rat. The effects are not as clear when disaggregated in this way but we agree that these figures add completeness.

Reviewer #3 (Recommendations for the authors):The central question in this study is whether value representations in the (rodent) OFC drive choice or whether they drive learning. The core question is an important one in neuroscience because the OFC is the focus of a great deal of long-standing debate. The authors of the study separate these possibilities using a well-known multi-step task. The task is powerful because it provides a computationally strong and elegant way to separate these things. Note that one innovative feature of the authors' research program is training rodents to perform this task – they have done so in previous work, and, here leverage that advance to begin to answer neuroscientific questions. Formally speaking, the authors use two complementary methods, unit recordings and optogenetic silencing. Results from both methods wind up pointing in the same direction, towards the learning account and away from the choice account.I have a few thoughts that relate more to the presentation than to the data. I wonder how the frame structure of learning vs. choice relates to the existing larger literature on OFC. From my understanding, this frame skips cognitive mapping functions and the related expectancy signaling accounts, and also skips accounts in which OFC helps navigate complex of model-based choices. Im not saying that these accounts are all in conflict or must be addressed, but most readers will be familiar with them, and the authors should work through how they all relate. Are they saying these others are all part of a more general learning-based account? Additional explanation would help.

We thank the reviewer for pointing this out. We believe our data are consistent with either of these accounts, while focusing on aspects that have not yet been the main focus of studies in those accounts. We have expanded our Discussion section to clarify this. The relevant paragraphs now read:

“Our data are also broadly consistent with the idea that the OFC represents a “cognitive map” of task-relevant state information. […] A promising direction for future work would be to perform experiments analogous to ours in the regions of the rodent brain that are plausibly homologous to ventral parts of the medial network (VO and MO) as well as elsewhere in PFC.”

And

“Our results are consistent with the view that the OFC carries expectancy signals (Schoenbaum and Roesch 2005; Rudebeck and Murray 2014) indicating which outcomes are expected to follow from the current state. […] Experiments using this task are therefore blind to the difference between representations of the expected probability of this reward (the water droplet) in particular and representations of expected value that abstract over many possible rewarding outcomes.”

Relatedly, it would seem that the null hypothesis here is that OFC does both learning and choice – why not?? Is there any a priori reason it shouldn't do them? It seems reductive to insist a brain region can only have one function.

We agree very much that it is plausible that the OFC participates in both learning and choosing (indeed we had expected that our study would show that this was the case in our task). We have adjusted the wording in our abstract, introduction, and discussion to be more clear about this. Relevant sentences now read (emphasis added):

Abstract:

“Firstly, they drive choice: the expected values of available options are compared to one another, and the best option is selected. Secondly, they support learning: expected values are compared to rewards actually received, and future expectations are updated accordingly. Whether these different functions are mediated by different neural

representations remains an open question.”

Introduction:

“Recording studies in many species have revealed neural correlates of expected value in the OFC (Thorpe, Rolls and Maddison, 1983; Schoenbaum, Chiba and Gallagher, 1998; Gottfried, O’Doherty and Dolan, 2003; Padoa-Schioppa and Assad, 2006; Sul et al., 2010), but it has not been clear whether these neural correlates of expected value are selective for roles in learning and in choosing. This, along with limitations inherent in neural perturbation studies, has made it difficult to determine whether the OFC plays a role in learning, choosing, or both.”

“We asked whether OFC neuron firing rates were correlated with the expected values of the items being learned about, the values being chosen between, or both.”

“To causally probe whether OFC plays a role in learning, in choosing, or in both processes, we transiently silenced OFC activity, and found that the pattern of behavior induced by this silencing was reproduced in our computational model only when we disrupted only the role of expected value in learning. Disrupting the role of value in choice did not reproduce this effect.”

Results:

“We sought to determine whether our neural recording data contained correlates of choice-related value signals, outcome-related value signals, or both.”

Discussion:

“Value representations in the OFC have been reported in many tasks and species, but it is still unclear whether they drive one process, the other process, or both.”

So for example, the authors find that OFC is more active for outcomes than for offers. But it is still active for offers. One might therefore conclude most parsimoniously that OFC is involved more strongly in learning, but still involved in choice as well. Indeed, I would look at the significant CPDs for choice variables and say the authors have presented clear evidence for both. It's not even clear that bigger CPDs for outcome than offer implies that OFC is more involved in learning than choice – there are several reasons you could imagine why you would see this effect (e.g. learning is more difficult, requires more attention, is more narrowly focused on OFC). These other possibilities are all consistent with the "both" theory, which the authors do not seem to consider. Like, these sentences, for example, seem to contain within them the assumption that weak but significant correlations are otiose… which seems unlikely…. "We find that neurons in the OFC correlate with the expected values of the items being learned about, but only weakly with the expected values of the items being chosen between. This indicates that, within a repeated-trials task, neural representations in the OFC carry value information that is selective for a role in learning, but do not seem to carry value information selective for a role in choice." Moreover, the analysis is conservative – if I understand correctly it attributes all variance to other confounding variables and counts only remaining variance. I see why the authors would do it, but this would seem to artificially deflate responses. That itself might not be bad, if we keep it in mind when interpreting the data. But the result here is that the effects are smaller than expected.Even so, the effect size numerically is not as big as the authors' rhetoric implies – 1.1% is the good number and 0.52% is the bad number. Is it so obvious that we should interpret these as the authors want us to – that 1.1% means OFC is all about learning and 0.52% means it has nothing to do with choice? Stronger results are reported a few sentences later (the ratio is 3.8 or 4.0). However, again, I think it is a step too far to consider this to be evidence that OFC does one thing and not another. I am reminded, for example, of the fact that nodulation by attention in visual area V4 and strong driving by visual stimuli – in those areas, the modulation ratio is about 5:1. That doesn't mean, however, that V4 doesn't participate in attention.

We revised the text to be clear that the electrophysiology results do show significant, if weaker, correlates of choice-related value information, and that the electrophysiology data does not rule out the idea that these units may play some role in choice. Relevant passages now read:

Introduction

“This indicates that, within a repeated-trials task, neural representations in the OFC carry value information that is largely selective for a role in learning, but only weakly carry value information selective for a role in choice.”

Results

“Across all of these variants a clear pattern was present: neural activity in OFC encodes the expected value of the visited outcome port more strongly than it encodes either type of value information about the choice ports. In computational models (Figure 2a), this type of value information plays a role in supporting learning, but does not play a direct role in choice. Our neural recording results therefore suggest that the OFC may play a role in supporting learning, but cast doubt on the idea that they play a strong role in driving choice directly.”

Discussion

“we find weak (though significant) representation of values associated with the available choices (“choice values”), little or no effect of silencing OFC at the putative time of choice, and effects of silencing inconsistent with impairing choice values in a computational model. Instead, we find strong representation of values associated with immediately impending reward outcomes (“outcome values”), a strong behavioral effect of silencing OFC at the time of those representations, and effects of silencing that are consistent with specifically impairing the use of outcome values for learning.”

We have also modified the main text CPD timecourse analysis to subtract from each regressor’s CPD the mean of the values of that CPD found in the permuted datasets (Figure 4b). This correction makes these numbers easier to interpret, as the null hypothesis predicts a value of zero for each, rather than some small positive number as in the previous version of our analysis. The outcome value regressor reaches a peak value of 0.82, while the choice and chosen value regressors reach peak values of 0.21 and 0.29.

For the causal manipulation, I think a clearer case can be made about its novelty. The issue is a series of papers by Gardner and Schoenbaum using causal/optogenetic approaches to demonstrate the relevance of OFC to learning and learning-like processes. The authors here say that "Our results build on this by pinpointing, out of all those possibilities, a specific computational role for OFC: it provides outcome value estimates to a learning system, elsewhere in the brain, that updates the estimates of these values." However, again, this claim is not as strong as it seems – there is no reason given to think that OFC only has one role or that its role in this task is the only role that it plays in cognition more generally. Reading this paper along with the Gardner papers, I would say that the authors' conclusion is wrong in that they do not reject alternative roles for OFC other than choice.

We agree that the Gardner and Schoenbaum papers demonstrate a role for the OFC in updating choice mechanisms. We believe that our data are consistent with this view, but add something more specific. We have revised our Discussion section to be more clear about this. The relevant passage now reads:

“[The Gardner and Schoenbaum] results indicate that the OFC activity is crucial specifically at times when a behavioral strategy is being updated. They leave open the question of what computational role this activity plays. They are consistent with several possible roles, including identifying when the strategy should be updated, driving the update of that strategy directly, or driving decisions between recently-updated alternatives. Our analysis of the coarse effects of inactivation in different time windows replicates this result. Our analysis of the fine-grained effects of inactivation builds on it, using a cognitive model to jointly interpret neural recordings and trial-by-trial behavioral effects of inactivation. They indicate that the detailed effects of inactivation are attributable in particular to degradation of the outcome value signal (and not, for example, the reward signal, which is stronger and carried by a larger fraction of clusters). This suggests that, at least in our task, the OFC plays a particular computational role in supporting update: it provides outcome value estimates to a learning system, elsewhere in the brain, that updates the behavioral policy.”